# Choice-relevant information transformation along a ventrodorsal axis in the medial prefrontal cortex

David J.-N. Maisson [1,2,3,4 ✉], Tyler V. Cash-Padgett[1,2,3,4], Maya Z. Wang[1,2,3,4], Benjamin Y. Hayden[1,2,3,4], Sarah R. Heilbronner[1,2,3,4] & Jan Zimmermann [1,2,3,4]

Choice-relevant brain regions in prefrontal cortex may progressively transform information about options into choices. Here, we examine responses of neurons in four regions of the medial prefrontal cortex as macaques performed two-option risky choices. All four regions encode economic variables in similar proportions and show similar putative signatures of key choice-related computations. We provide evidence to support a gradient of function that proceeds from areas 14 to 25 to 32 to 24. Specifically, we show that decodability of twelve distinct task variables increases along that path, consistent with the idea that regions that are higher in the anatomical hierarchy make choice-relevant variables more separable. We also show progressively longer intrinsic timescales in the same series. Together these results highlight the importance of the medial wall in choice, endorse a specific gradient-based organization, and argue against a modular functional neuroanatomy of choice.

---

[1] Department of Neuroscience, University of Minnesota, Minneapolis, MN, USA. [2] Center for Magnetic Resonance Research, University of Minnesota, Minneapolis, MN, USA. [3] Center for Neuroengineering, University of Minnesota, Minneapolis, MN, USA. [4] Department of Biomedical Engineering, University of Minnesota, Minneapolis, MN, USA. ✉email: maiss002@umn.edu

E conomic choice is mediated by a large number of brain regions, including several in the prefrontal cortex[1–3]. The principles delineating the allocation of specific choice-related functions to specific regions remain unclear. On one hand, modular theories hold that specific brain regions can be associated with particular nameable functions, such as evaluation, comparison, and action selection[1,4–6]. On the other hand, distributed approaches to understanding choice hold that particular elements of choice do not correspond neatly to anatomical regions[7–10]. These distributed approaches are inspired by classic connectionist theories as well as by modern deep learning approaches[9,11–13]. They are also inspired by cognitive and philosophical theories of distributed cognition[14–16], and by analogy to the form vision system, where gradient-based models have come to replace historically dominant modular models[10,17].

Within the domain of distributed models, early mass-action and equipotentiality models have lately been supplanted by hierarchical distributed models[3,18,19]. We and others have proposed a related but conceptually distinct approach defined by a gradual transformation of information[9,10,20,21]. For example, specific circuits within the prefrontal cortex may be organized into a gradient, so that each anatomical region implements part of a transformation of task-relevant information from a domain of options to a domain of actions[10]. We have proposed that each region untangles information about the best action, which is latent in the early representations and which, through serial processing, is transformed into appropriate choice actions[10]. This view is inspired by and is an extension of, modern untangling-based models of form vision[17].

A critical prediction of gradient-based models is that it should be possible to arrange medial prefrontal regions into a series on the basis of their functional properties. Discussions of prefrontal gradients have typically focused on the lateral surface, or, when examining the medial wall, on the rostrocaudal axis[22–28]. However, even in the lateral prefrontal cortex, there is evidence of shared functionality between distinct subregions. Differential contributions appear to be quantitative and indicative of a graded flow of information[29]. We were interested, instead, in the ventrodorsal dimension of the medial prefrontal cortex. Neuroeconomists have long proposed that the orbitofrontal and ventromedial prefrontal cortices serve as the entry point of choice-relevant sensory information into the prefrontal cortex and that the motor cortex serves as the output[5,30–32]. The medial wall inferior to the premotor cortex, which includes areas 14, 25, 32, and dorsal anterior cingulate cortex (dACC), which we call area 24, looks to be a likely pathway linking OFC to pre/motor areas. These regions also have prominent limbic, visceral, and reward-related functions, suggesting they may contribute to valuation and perhaps to choice as well[33–35].

However, there are several possible functional gradients that are consistent with known anatomy. First, it could follow topology in a rough ventrodorsal direction (14→25→32→24). Second, it could match the contour of the genu of the corpus callosum (25→14→32→24). Third, cytoarchitecture suggests that the less differentiated cingulate areas (25, 32, and 24) may precede the more differentiated area 14[36]. Other cytoarchitectural studies suggest that pre- and subgenual regions (14, 25, and 32) may have shared functions but differ qualitatively from the postgenual 24[37]. These specific pathways have not, to our knowledge, been functionally evaluated. Despite this, identifying the functional gradient, if one exists, is critical for establishing the neuroscience of economic choice[1].

We examined a composite dataset consisting of single-unit responses collected in these four brain regions. We included previously published data for 14, 25, and 24, and newly collected data for area 32[38–40]. Instead of looking for specific functions that would distinguish these regions from each other, we took a function-first approach: we selected key putative signatures of participation in specific elements of choice before beginning the study and then characterized each region in these functions.

Here, we show two major results. First, the regions all show signatures of all tested functions and do not differ qualitatively in whether they carry certain information or have signatures of choice processes. Second, a decoding analysis shows both stronger decodability on twelve dimensions consistent with a single gradient, specifically one that progresses from 14→25→32→24 (that is, that follows a ventrodorsal direction). This second result is complemented by a demonstration that intrinsic timescale shows the same pattern. Together these results support a specific ventrodorsal medial prefrontal gradient and, simultaneously, argue against a modular view in which conceptually distinct functions are reified in neuroanatomy.

## Results

**Behavioral results**. Rhesus macaques (Macaca mulatta) performed one of two structurally similar two-alternative forced-choice gambling tasks[38,39] (see Methods). Briefly, subjects chose between two risky options presented asynchronously (Fig. 1A, B). After the second offer was presented, a go cue indicated that the subject was free to shift gaze toward the target option to indicate a choice. Thus, the task was naturally divided into three epochs, corresponding to the periods immediately following offer 1, offer 2, and choice. The offer 2 epoch was the first during which the subject could compare subjective values and select an action. Note, however, that subjects can and likely do form tentative partial choices based solely on the value of the first offer[39].

Behavior in these tasks has been explored at length and is not reanalyzed here (the most detailed analyses are published in ref. [41]). Briefly, behavior reflected an understanding of all important task variables with very weak order or side biases. We defined the expected value of an offer as the product of the offer magnitude (in µL juice) and the probability of reward. Thus, for a basic characterization of behavior, we computed the frequency with which a given offer was chosen when it had a higher expected value. We determined the proportion of trials on which the subject chose the first offer and we compared it to the difference in expected values of the two offers for each trial. Subjects' behavior described a sigmoidal function (Fig. 1C). Subjects most frequently chose the offer with the higher expected value (ventromedial prefrontal cortex (vmPFC) sessions: 84.55% of trials; subgenual ACC (sgACC) sessions: 78.62%; pregenual ACC (pgACC) sessions: 74.97%; dACC sessions: 75.57%; $p < 0.0001$ in all cases; 1-sample $t$ test), consistent with the idea that they had a basic understanding of the task.

**Firing rates in all regions encode values of offers**. We recorded neuronal activity from four brain regions: 156 neurons (106 from subject B and 50 from subject H) in vmPFC area 14, 146 neurons (77 from subject B and 69 from subject J) in sgACC area 25, 213 neurons (110 from subject B and 103 from subject V) in pgACC area 32, and 129 neurons (55 from subject B and 74 from subject J) in dACC area 24. The data from pgACC have not been previously published. Some data from vmPFC, sgACC, and dACC have been previously published, although the key analyses here have not previously been reported[38–40,42–44]. We collected these recordings from 4 subjects (B, H, V & J see Methods and Fig. 1D). For each area, we recorded from two subjects, although we did not use the same subjects for all areas. We did not observe marked behavioral differences across subjects and therefore did not expect nor observe qualitative differences between subjects. We collected data from dACC and sgACC in the token risky choice task; we collected data

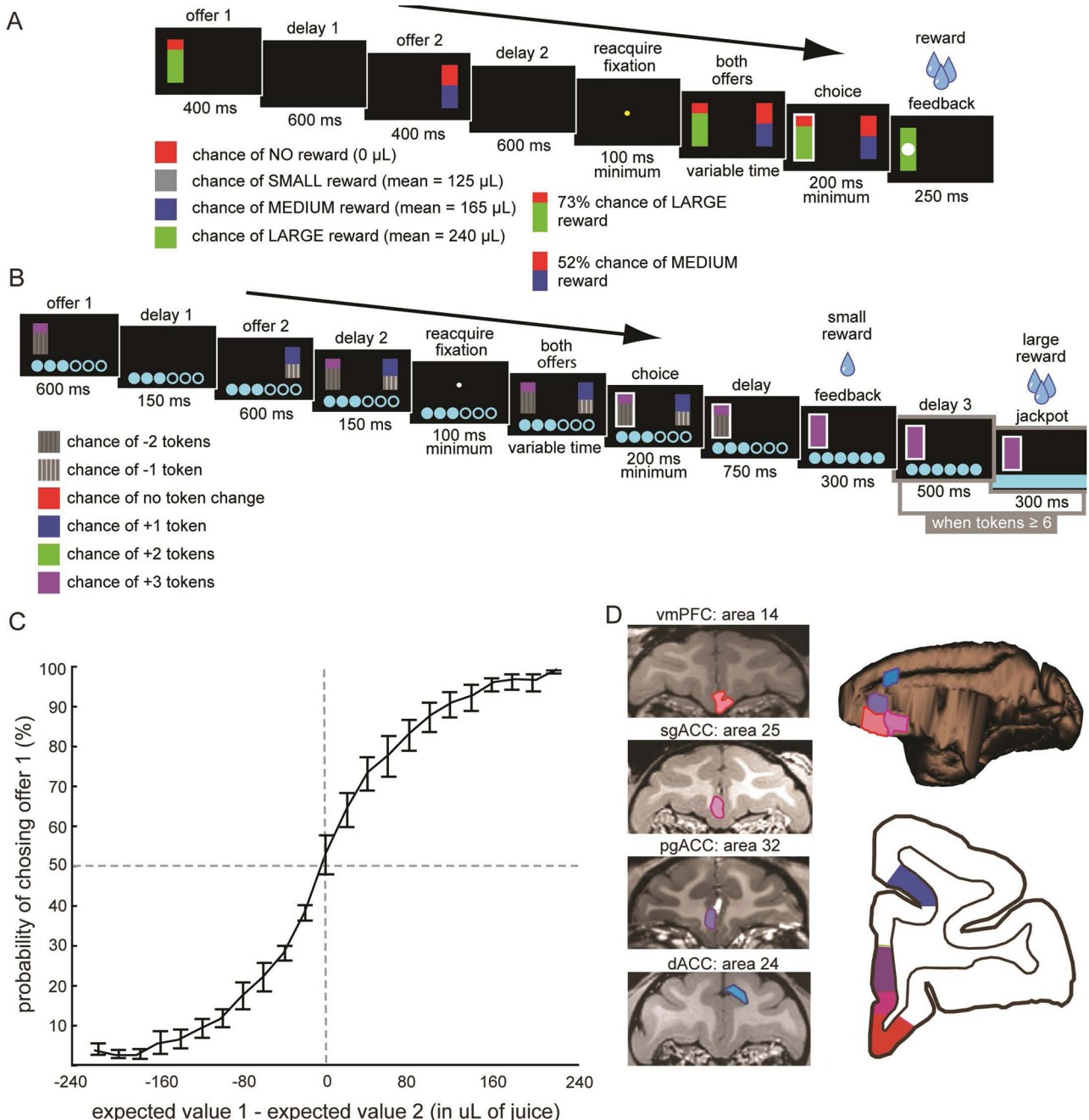

**Fig. 1 Tasks, behavior, and structures. A** Risky choice task: the first offer is presented, followed by a delay period, after which the second offer is presented. After another delay period, fixation is reacquired for a minimum of 100 ms. Both offers are then presented, a choice is made, and the choice is probabilistically rewarded. Offers consist of a reward magnitude (color of the non-red portion of the bar) and a probability (size of the colored portion). **B** Token risky choice task: equivalent format but the reward is tokenized. Once tokens reach 6, a reward is delivered. **C** The sigmoid is plotted along with the mean choice behavior across all subjects and recording sessions ($n = 4$ animals recorded across 525 sessions). Error bars reflect the standard error across sessions. Source data are provided as a Source Data file. **D** MRI scans of the brain areas targeted for recordings (red = vmPFC, pink = sgACC, purple = pgACC, blue = dACC). A single representative subject is chosen for each area even though two subjects were recorded for each. The right panels denote the targeted brain areas on both transverse and coronal planes.

from vmPFC and pgACC in the risky choice task (Fig. 1A, B). The basic format for each of the tasks during the selected time period for analysis, from within each trial, is essentially the same. We do not believe the small differences between the two tasks influenced the results we present here.

**Neuronal activity is modulated by offer value.** We examined the proportion of neurons in each region selective for the value of

offer 1 during epoch 1. Figure 2B–E shows the average firing rate responses of an example neuron from each of the four regions. In all four regions, a significant proportion of neurons encoded the value of offer 1 during the offer 1 epoch (vmPFC: 16.03%; sgACC: 10.27%; pgACC: 10.98%; dACC: 26.36%; $p < 0.01$ in all areas, binomial test). Likewise, neurons in three of the four regions encoded the value of offer 2 during epoch 2 (vmPFC: 14.1%; sgACC: 7.53%; pgACC: 10.98%; dACC: 14.73%; $p < 0.01$ in all

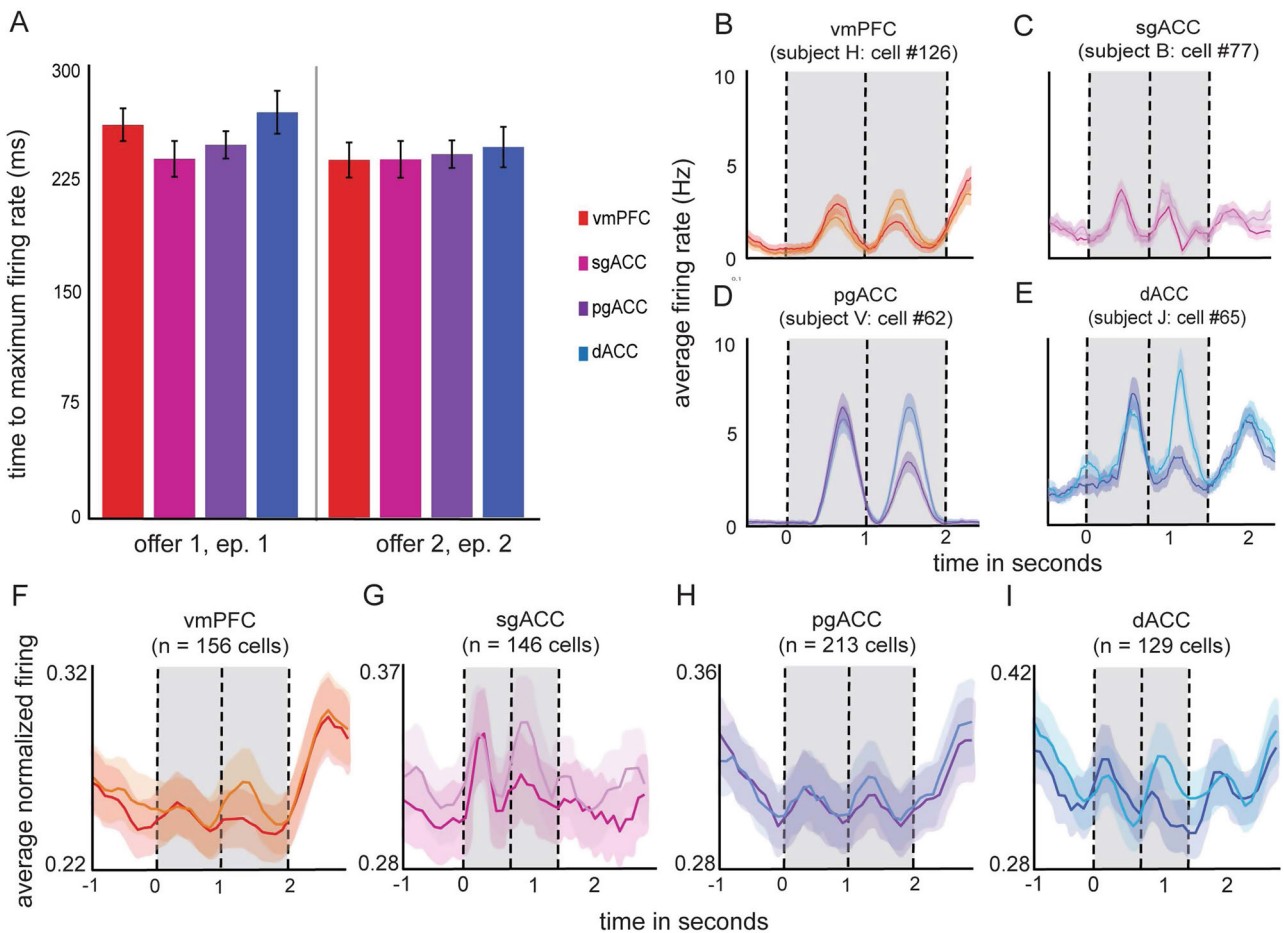

**Fig. 2 Response latency and traces of average firing rates across all trials drawn from responses of single sample neurons. A** Bars plot the average latency to maximal firing rates in response to the onset of both offer 1 and offer 2 (red = vmPFC: $n = 156$ neurons; pink = sgACC: $n = 146$ neurons; purple = pgACC: $n = 213$ neurons; blue = dACC: $n = 129$ neurons). Error bars indicate the standard error across trials. Source data are provided as a Source Data file. **B–E** Peri-stimulus time histogram responses of sample neurons with firing rates that are significantly correlated with the expected values of both offers. Traces are grouped by trials on which the value of either offer 1 (darker color) or offer 2 (lighter color) was larger. Traces are smoothed, for display, with a 200 ms sliding boxcar. Each trace (a measure of center) indicates the average firing rate for a given neuron across trials (B: $n = 711$ trials; C: $n = 425$ trials; D: $n = 766$ trials; E: $n = 429$ trials). Average firing rates are computed in spikes per second. The onset of offer 1 is set to time 0. The vertical lines indicate the start times of periods in the trial (the onset of offer 1, offer 2, and fixation). Error ribbons denote standard error. Source data are provided as a Source Data file. **F–I** Same as **B–E**, except that these are the time-resolved population averages. Each trace (a measure of center) indicates the average firing rate across neurons. Traces are smoothed with a 200 ms sliding boxcar for display purposes. Error ribbons denote standard error. Source data are provided as a Source Data file.

areas except sgACC (trending at $p = 0.063$, binomial test). Finally, neurons encoded the value of offer 1 during epoch 2 (i.e., presumably, working memory for value, vmPFC: 9.62%; sgACC: 10.27%; pgACC: 9.41%; dACC: 14.73%; $p < 0.01$ in all areas, binomial test; Fig. 3). We also plotted the average explained variance ($r^2$ computed from the correlation analyses used to demonstrate neuronal modulation by offer value), for each region across time.

**Putative signatures of the choice process are found in all four regions.** We next asked whether each brain region contains a value signal that reflects the integration of the two features that determine value: probability and magnitude[44]. For each brain region, we computed regression weights for each neuron's normalized ($z$-scored) firing rates for the two variables. We then examined how those variables related to each other across the population. A positive correlation between regression coefficients indicates that both offer features are encoded using a correlated coding scheme. In other words, it indicates that the population of

neurons has thrown out information about the details of the components and has begun to compute an integrated value signal[44]. We observed a significant positive relationship in all regions ($p < 0.001$ in all regions; Fig. 4A). This result indicates that feature integration is not a unique feature of any region, but instead, is broadly shared across the medial PFC.

When attention shifts from one option to another, it is possible that the same population of neurons encodes the new option in the same manner as it encoded the first one, like neurons in the visual cortex do for visual stimuli[45–47]. In other words, neurons may act as a flexible filter for value; we have called this principle attentional alignment (as opposed to a labeled line code for value[46]). We next asked whether the attentional alignment is a principle shared in all four of our regions. To do so, we regressed normalized firing rates from epoch 1 onto the expected value of offer 1 and regressed normalized firing rates from epoch 2 on the expected value of offer 2. We then correlated these resulting coefficients. A positive correlation is evidence for attentional alignment. All four regions exhibited significant ($p < 0.01$) positive correlations (Fig. 4B).

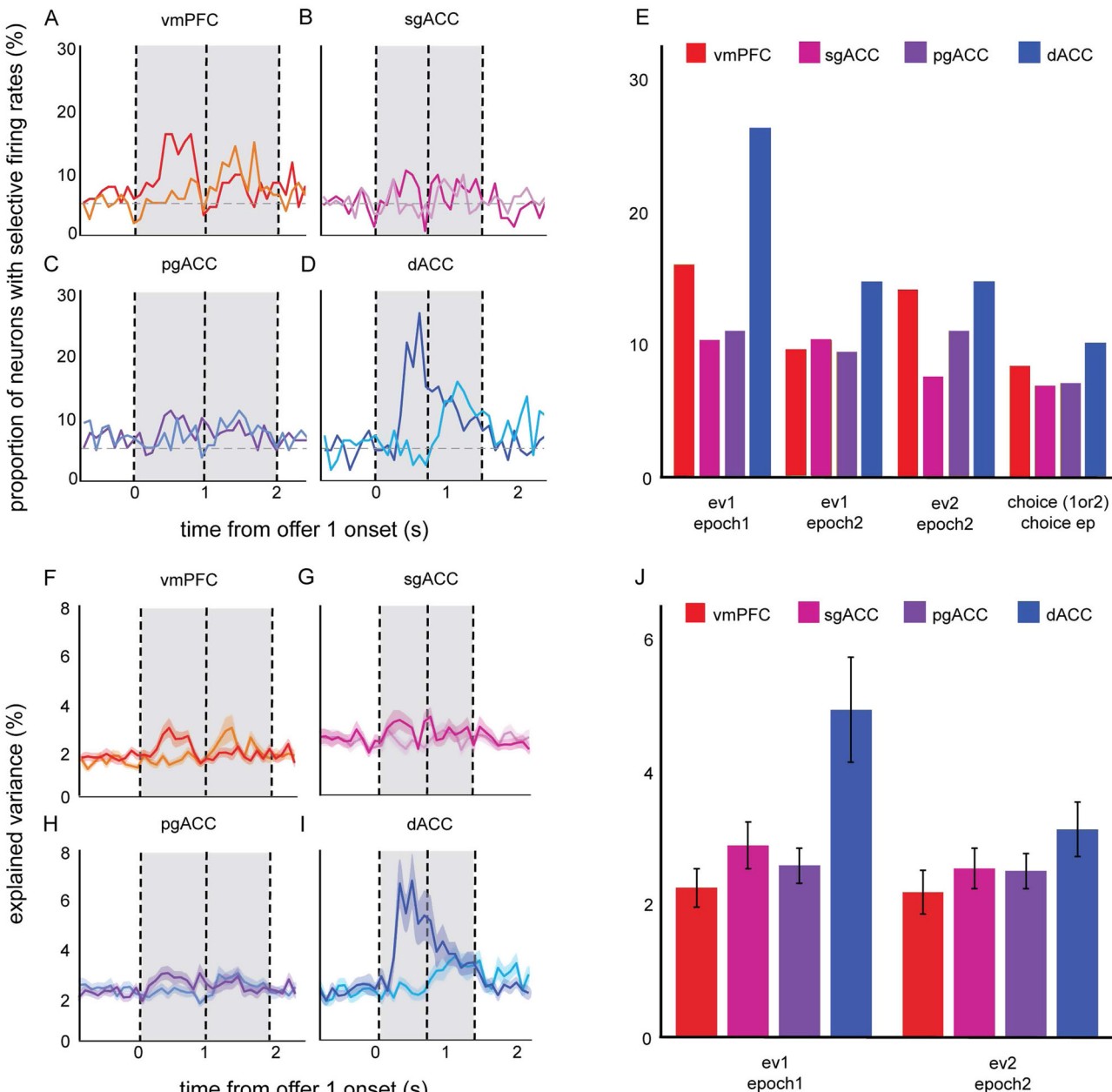

**Fig. 3 Selectivity of firing rates to offers and choice. A–D** The proportion of neurons that have firing rates that are correlated with the expected value of offer 1 (darker color) and offer 2 (lighter color) across a 3-s period during the trial (red = vmPFC, pink = sgACC, purple = pgACC, blue = dACC). The onset of offer 1 is set to time 0. The vertical lines indicate the onsets of offer 1 and offer 2. Source data are provided as a Source Data file. **E** A summary of the proportion of neurons selective to offers and choice, within given epochs. Each bar indicates the proportion within a given brain area. Source data are provided as a Source Data file. **F–I** The average variance explained ($r^2$) across neurons, from correlating firing rates with the expected values of offer 1 (darker color) and offer 2 (lighter color). Each trace (a measure of center) indicates the average explained variance across neurons. Vertical lines and shading on the x-axis indicate trial events. Error ribbons denote standard error. Source data are provided as a Source Data file. **J** A summary of average explained variance, collapsed across the respective offer epoch (500 ms window), relative to each of the offers (vmPFC: $n = 156$ neurons; sgACC: $n = 146$ neurons; pgACC: $n = 213$ neurons; dACC: $n = 129$ neurons). Error bars denote the standard error across neurons. Source data are provided as a Source Data file.

We next asked if there was evidence for direct comparison between offers by the principle of mutual inhibition[38,39]. We regressed normalized firing rates from epoch 2 on offer 1 onto responses from epoch 2 on offer 2. If the encoding for value 1 and value 2 during the same epoch are anti-correlated, then the encoding of one value comes at the expense of the other. It is an indication of the direct comparison of offers and thus a signal of choice. All recorded regions exhibited significant ($p < 0.05$) negative correlations (Fig. 4C) between regression weights. This

result indicates that all four regions show evidence of value comparison via mutual inhibition.

**Overlapping neuronal populations.** Next, we asked if there were distinct or shared populations of neurons encoding the previously described functions. We repeated the integration, alignment, and inhibition analyses (i.e., the correlations of select regression weights), but now with absolute (i.e., unsigned) values of the

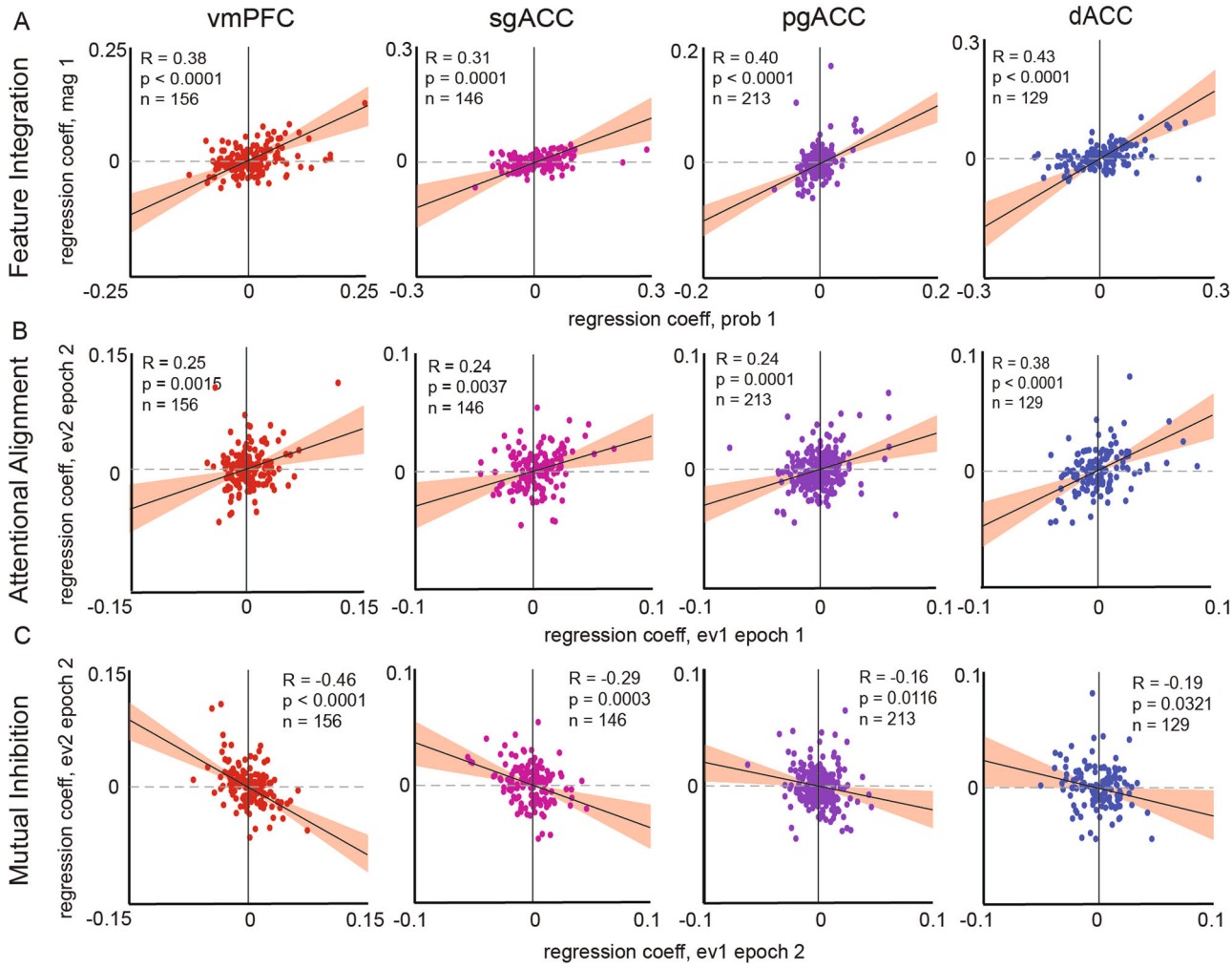

**Fig. 4 Economic choice functions. A** Feature integration. From left to right: Scatter plots of regression coefficients from regressing normalized epoch 1 firing rates on the probability of offering 1, against the regression coefficients from the magnitude of offer 1 (red = vmPFC, pink = sgACC, purple = pgACC, blue = dACC). The diagonal black line indicates the slope of the correlation between regression coefficients (Pearson's correlation, with *p*-value computed as a two-sided test). The red ribbons indicate the 95% confidence intervals. Source data are provided as a Source Data file. **B** Attentional alignment. Scatter plots are of coefficients from regressing normalized epoch 1 firing rates on the expected value of offer 1, against the regression coefficients from epoch 2 firing rates on the expected value of the offer. The diagonal black line indicates the slope of the correlation between regression coefficients (Pearson's correlation, with *p*-value computed as a two-sided test). The red ribbons indicate the 95% confidence intervals. Source data are provided as a Source Data file. **C** Mutual inhibition. Scatter plots are of coefficients from regressing normalized epoch 2 firing rates on the expected value of offer 1, against the regression coefficients from epoch 2 firing rates on the expected value of offer 2. The diagonal black line indicates the slope of the correlation between regression coefficients (Pearson's correlation, with *p*-value computed as a two-sided test). The red ribbons indicate the 95% confidence intervals. Source data are provided as a Source Data file.

regression weights. We have previously shown that such correlations indicate shared or overlapping functional populations[48]. If the (unsigned) strength of encoding of one offer (or feature) is positively correlated with the degree of encoding of the other offer, then the populations associated with encoding the two variables overlap more than expected by chance. We found a significant positive correlation (vmPFC: $r = 0.31$; sgACC: $r = 0.31$; pgACC: $r = 0.44$; dACC: $r = 0.41$; $p < 0.001$, all areas, Pearson's correlation) between unsigned regression coefficients for the integration function (epoch 1 firing rates regressed on offer 1 magnitude and on offer 1 probability). We also found a significant positive correlation between unsigned regression coefficients for all areas (vmPFC: $r = 0.34$; sgACC: $r = 0.23$; pgACC: $r = 0.33$; $p < 0.01$, Pearson's correlation), except for dACC ($r = 0.17$, $p = 0.058$, Pearson's correlation), representing alignment (epoch 1 firing rates regressed on expected value 1 and epoch 2 regressed on expected value 2). Finally, vmPFC ($r = 0.29$)

and pgACC ($r = 0.25$) showed significantly positive correlations ($p < 0.001$ in both cases, Pearson's correlation) between unsigned regression coefficients for inhibition (epoch 2 firing rates on the expected value of each offer). These results indicate that, in most areas and across functions, encoding is mostly supported by the same, or at least overlapping, sets of neurons.

**Intrinsic timescales are longest at the top of the anatomical gradient**. The results presented so far demonstrate broadly overlapping functions across regions. We next asked whether there is evidence of a functional gradient. We first considered intrinsic timescales. Intrinsic timescales are a population-level statistic describing fluctuations in a neuronal signal that are agnostic to the task and corresponding variables[32]. Murray and colleagues[32] proposed that intrinsic fluctuations are a function of increased modulatory strength. They suggested that increased modulation is due to increased recurrent network activity, which

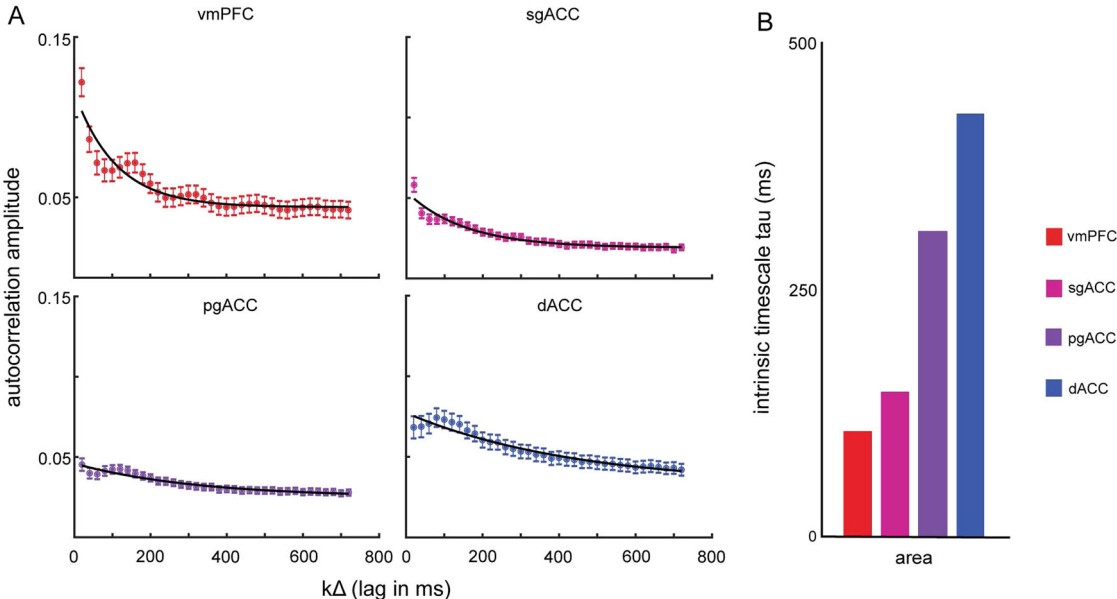

**Fig. 5 Intrinsic timescales. A** Spike-count autocorrelations calculated at increasing lags for each of the four brain areas (red = vmPFC: $n = 156$ neurons; pink = sgACC: $n = 146$ neurons; purple = pgACC: $n = 213$ neurons; blue = dACC: $n = 129$ neurons). The decay of autocorrelation amplitude with lag was fitted by an exponential decay function with offset (black line). Error bars indicate the standard error for autocorrelation amplitude across neurons at a given lag. Source data are provided as a Source Data file. **B** A summary table of the intrinsic timescales, extracted from the exponential decay function. The figure demonstrates a smooth increase of intrinsic timescale along a gradient. Source data are provided as a Source Data file.

in turn increases along an anatomical gradient. Thus, longer intrinsic timescales would be indicative of increased modulatory strength and, therefore, a higher position along the gradient.

We estimated and compared the intrinsic timescales of each recorded region from a temporal decay function. The neural activity used to estimate the intrinsic timescales came from the last two seconds of the inter-trial interval, that is, before the onset of the first offer on the next trial (similar to ref. [32]). This period is absent of any cues or information about either the previous or pending trial. We used the decay function to fit the autocorrelation of pre-trial spike data across a range of lags (see Methods). We found an increase of intrinsic timescale that seemed to map a medial prefrontal gradient onto a rough ventrodorsal gradient (vmPFC: 109.8 ms; sgACC: 152.76 ms; pgACC: 321.85 ms; dACC: 446.51 ms; Fig. 5). We confirmed a positive monotonic relationship by correlating the intrinsic timescales with the observed order (1–4) across the areas. The results showed a significant positive correlation between increasing order across the four areas, and the increasing intrinsic timescale ($r = 0.98$, $p = 0.022$, Pearson's correlation).

**Decoding accuracy supports a clear functional gradient.** We hypothesized that the accuracy with which expected value and choice can be decoded from firing rate patterns should increase along the anatomical gradient. To test this hypothesis, we trained (and cross-validated) a linear classifier to decode ten binary labels (specifically: high/low expected values for offer 1 and for offer 2, the difference between expected values of the two offers, offer position, choice, chosen offer position, chosen offer value, unchosen offer value, the choice on the previous trial, and reward on the previous trial) from firing rates. We investigated decoding accuracy for two of these variables (expected value of offer 1 and chosen offer position) in two different epochs, thereby yielding a total of 12 independent decoding analyses (see Methods).

We first looked at how accurately the offer 1 value could be decoded from firing rates in epoch 1 (Fig. 6A). The classifier decoded whether offer 1 on each trial had greater or lesser value

than the mean offer value across trials significantly better than chance (binomial test, $p < 0.0001$) for all regions: vmPFC (61.4%), sgACC (63.1%), pgACC (69.8%), and dACC (73.7%). Notably, when we compared the proposed gradient order with the decoding accuracy distributions, decoding accuracy increased with gradient order ($\rho = 0.81$, $p < 0.001$, Spearman's correlation). The same principle applied to offer 1 value in epoch 2 (a putative signature of working memory for value[49]; Fig. 6B). The classifier decoded whether offer 1 on each trial had greater or lesser value than the mean offer value (binomial test, $p < 0.001$) for all regions: vmPFC (57.7%), sgACC (60.1%), pgACC (68.5%), and dACC (69.4%). As above, decoding accuracy increased with gradient order ($\rho = 0.69$, $p < 0.001$, Spearman's correlation).

We next looked at how accurately offer 2 values (greater or less than mean) could be decoded from firing rates in epoch 2 (Fig. 6C). As above, the proposed gradient order (binomial test, $p < 0.0001$) for all regions: vmPFC (62.6%), sgACC (64.9%), pgACC (69.2%), and dACC (66.6%), accuracy increased with gradient order ($\rho = 0.24$, $p < 0.001$, Spearman's correlation). The same pattern is observed with the difference between offer values (greater or less than mean) in epoch 2 (Fig. 6D, binomial test, $p < 0.001$) for all regions: vmPFC (54.9%), sgACC (62.3%), pgACC (67%), and dACC (70.2%). Decoding accuracy increased with gradient order ($\rho = 0.82$, $p < 0.001$, Spearman's correlation).

Next, we looked at how accurately the chosen offer (offer 1 or 2) could be decoded from firing rates in the choice epoch (Fig. 6E). Note that this variable is orthogonal to the offer side since we observed essentially no spatial biases in choice. The classifier decoded choice on each trial (binomial test, $p < 0.0001$) for all regions: vmPFC (64.2%), sgACC (70.5%), pgACC (70.6%), and dACC (72.5%). Decoding accuracy increased with gradient order ($\rho = 0.77$, $p < 0.001$, Spearman's correlation). The same pattern is observed with the expected value of the chosen offer (Fig. 6F, binomial test, $p < 0.0001$) for all regions: vmPFC (66.1%), sgACC (69.1%), pgACC (69.8%), and dACC (70.8%). Decoding accuracy increased with gradient order ($\rho = 0.59$, $p < 0.001$, Spearman's correlation). This pattern was also observed with

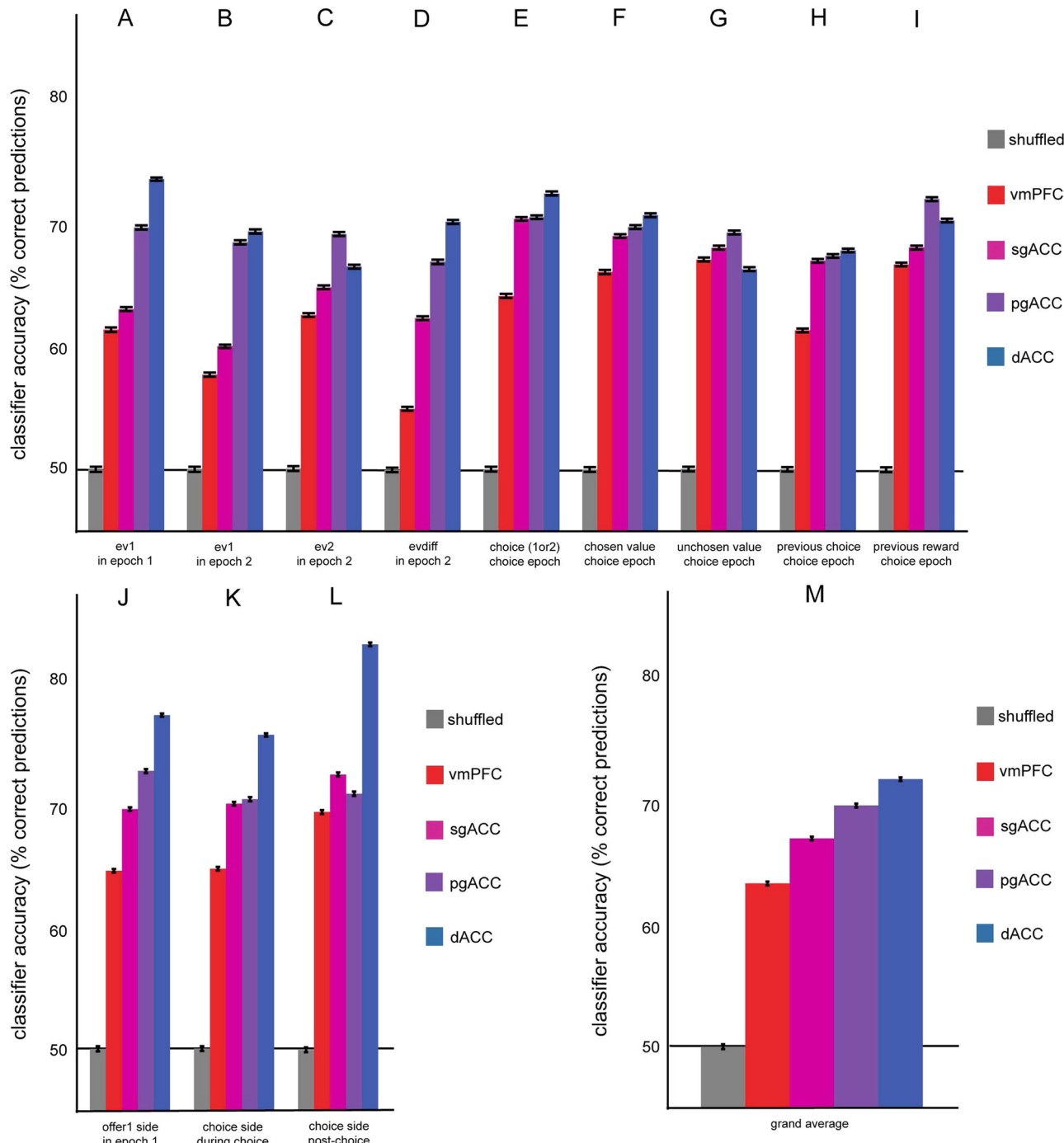

**Fig. 6 Decoding analysis. A–M** Summary of classification accuracies for each of the given labels (red = vmPFC, pink = sgACC, purple = pgACC, blue = dACC; $n = 1000$ bootstrapped samples). Bars plot the average decoding accuracy across cross-validation iterations. In a given epoch, a linear classifier was trained to identify the value of a binary label (indicated on the x-axis). The accuracy of the trained model was tested by cross-validation to predict the label value on untrained data (permutation average indicated on the y-axis). The accuracy of a model trained on randomly shuffled data is indicated by the gray bar. Error bars represent the standard error over the variance across cross-validations. Source data are provided as a Source Data file.

the expected value of the unchosen offer (offer 1 or 2, Fig. 6G, binomial test, $p < 0.0001$) for all regions: vmPFC (67.1%), sgACC (68.1%), pgACC (69.3%), and dACC (66.4%). Although the effect was nearly significant, decoding accuracy did not significantly increase with gradient order ($\rho = -0.03$, $p = 0.058$, Spearman's correlation).

Finally, we looked at how accurately the chosen offer (offer 1 or 2) from the previous trial could be decoded from firing rates in the current choice epoch (Fig. 6H). The classifier decoded choice

on each trial (binomial test, $p < 0.0001$) for all regions: vmPFC (61.3%), sgACC (67.1%), pgACC (67.4%), and dACC (67.9%). Decoding accuracy increased with gradient order ($\rho = 0.51$, $p < 0.001$, Spearman's correlation). The same pattern was observed with experienced reward (rewarded or not rewarded) from the previous trial (Fig. 6I, binomial test, $p < 0.0001$) for all regions: vmPFC (66.7%), sgACC (68.1%), pgACC (72.1%), and dACC (70.3%). Decoding accuracy increased with gradient order ($\rho = 0.66$, $p < 0.001$, Spearman's correlation).

**Graded organization of coding of spatial information**. We looked at how accurately the position of the offer on the monitor could be decoded from firing rates in epoch 1 (Fig. 6J). The classifier decoded whether offer 1 on each trial was on the right or left (binomial test, $p < 0.0001$) for all regions: vmPFC (64.5%), sgACC (69.5%), pgACC (72.6%), and dACC (77.2%). Decoding accuracy increased with gradient order ($\rho = 0.94$, $p < 0.001$, Spearman's correlation). We also looked at how accurately the position of the chosen offer on the monitor could be decoded from firing rates in the choice epoch (Fig. 6K). The classifier decoded whether the chosen side on each trial was on the right or left (binomial test, $p < 0.0001$) for all regions: vmPFC (64.7%), sgACC (70%), pgACC (70.4%), and dACC (75.6%). Decoding accuracy increased with gradient order ($\rho = 0.93$, $p < 0.001$, Spearman's correlation).

We also looked at how accurately the position of the chosen offer on the monitor could be decoded from firing rates in the post-choice epoch[50] (Fig. 6L). The classifier decoded whether the chosen side on each trial was on the right or left (binomial test, $p < 0.0001$) for all regions: vmPFC (69.3%), sgACC (72.4%), pgACC (70.8%), and dACC (83%). Decoding accuracy increased with gradient order ($\rho = 0.81$, $p < 0.001$, Spearman's correlation). Finally, we looked at the average accuracy across all 12 decoders (Fig. 6M). Average accuracy across all decoders was above chance (binomial test, $p < 0.0001$) for all regions: vmPFC (63.4%), sgACC (67.1%), pgACC (69.8%), and dACC (71.9%). Average decoding accuracy significantly increased with order ($\rho = 1$, $p = 0.017$, Spearman's correlation).

**Additional decoding controls reveal a consistent functional gradient**. We next asked whether the number of neurons collected in each region had some effect on measures of decodability, in case the different number of neurons collected for each region was contributing to our effects. We, therefore, decimated our datasets until each had the same number of cells (specifically, 125, a number chosen before analysis). We observed that for all of our twelve measures the order was unchanged, relative to the original results. Importantly, when we averaged the decoding accuracy across all decoders, we found a significant increase with gradient order. We conclude from this control analysis that our original findings are robust and do not depend on the number of cells in each sample.

Another possible confound comes from the fact that the expected value here is a continuous variable. If some structures have graded coding of the variable, a binary classifier may not be sensitive to this property. To address this concern, as an additional control, we performed a regression-based multinomial classifier for decoding multiple EV bins. To discretize EV, while still ensuring there would be a usable number of trials in each bin, we separated the expected value into six equally sized, consecutive bins. We separated trials based on the corresponding binned EVs and used these separated responses to construct pseudo-samples, following the same approach already described. We then trained the SVM and used the model to predict the offered and the chosen EV on an untrained cross-validation set. We found that the proposed gradient in the majority of our other findings (vmPFC->sgACC->pgACC->dACC) was also evident when we decoded these variables using multiple bins. As the results remain unchanged from the binary decoding of EV, we do not discuss them further.

**Baseline firing rates do not differ between structures**. First, we asked whether there were any differences in the intrinsic firing properties of neurons between the target structures. To measure baseline firing rates in each structure, we computed the average firing rate for each neuron during the 1-s pre-trial period, from −2 s to 0 s, where time 0 was set to the onset of offer 1. (Note that this is the same time window on which we performed our analysis of intrinsic timescales, see Methods). The baseline firing rate in vmPFC was 3.11 spikes/s. The baseline firing rate in sgACC was 3.69 spikes/s. The baseline firing rate in pgACC was 0.75 spikes/s. Finally, the baseline firing rate in dACC was 5.06 spikes/s. A Spearman correlation of these baseline firing rates with the proposed order was not significant ($\rho = 0.4$, $p = 0.75$, Spearman correlation). As the results are non-significant, we cannot draw any additional conclusions from this analysis.

Next, we wanted to determine if variability in the firing rates differed between structures. To measure firing rate variability, we computed the average Fano Factor (FF) across neurons in each region, replicating the procedure outlined in Chang et al.[51]. We segmented each 500 ms epoch into 100 ms bins. For each neuron, we calculate the variance and mean across trials within a given 100 ms bin. We then computed the FF as the variance divided by the mean and averaged this FF across neurons. Finally, in order to compare the FF across structures against the proposed gradient, we performed a Spearman correlation between the average FF across time bins and the expected gradient order. Across the time bins in epoch 1, the average FF in vmPFC was $1.93 \pm 0.199$ (SEM). The average FF in sgACC was $1.24 \pm 0.023$. The average FF in pgACC was $1.23 \pm 0.025$. The average FF in dACC was $1.35 \pm 0.042$. (Epoch 2, vmPFC: $1.92 \pm 0.171$; sgACC: $1.24 \pm 0.025$; pgACC: $1.23 \pm 0.024$; dACC: $1.32 \pm 0.042$). There was no significant correlation between gradient order and FF across the structure in either epoch 1 ($\rho = -0.4$, $p = 0.75$, Spearman correlation) or epoch 2 ($\rho = 0.4$, $p = 0.74$, Spearman correlation). As the results are nonsignificant, we cannot draw any additional conclusions from this analysis.

Lastly, we reasoned that qualitative encoding differences between structures would be associated with differences in the degree of dimensionality in neural activity between structures. To measure intrinsic dimensionality, we performed a principal component analysis on the normalized firing rates across neurons in each of the offer epochs, independently for each structure. For each neuron, we isolated the firing rates from both offer epochs. We calculated the mean firing rate across trials for each 20 ms time bin. Using these average responses, we composed a matrix of time X neurons. We performed an eigenvalue decomposition and computed the explained variance due to each of the first three principal components (PCs). We then calculated the degree of change, by calculating the slope, in explained variance from the first PC to the second PC, and again from the second PC to the third. We performed this same analysis for each offer epoch independently. Finally, we asked if the change in explained variance between the first three PCs was significantly correlated with the proposed order. We found that in epoch 1, the change in explained variance between PC1 and PC2 did not significantly change with gradient order ($\rho = -1$, $p = 0.083$, Spearman correlation). Similarly, the change in explained variance between PC2 and PC3, in epoch 1, did not significantly change with gradient order ($\rho = -1$, $p = 0.083$, Spearman correlation). We found that in epoch 2, the change in explained variance between PC1 and PC2 did not significantly change with gradient order ($\rho = -1$, $p = 0.083$, Spearman correlation). The change in explained variance between PC2 and PC3, in epoch 2, did not significantly decrease with gradient order ($\rho = -0.4$, $p = 0.75$, Spearman correlation). As the results are not significant, in any of these three analyses of intrinsic properties, we were unable to identify any qualitative differences in signatures of neural activity between the target structures.

**Offer encoding latency does not differ between areas**. First, we confirmed our hypothesis that there would be no differences

between areas in stimulus (i.e., the offer) response latencies. For both offer 1 and offer 2, we computed the latency of neural responses (see Methods). We defined latency as the time elapsed from the onset of the offer stimulus until firing rates in the respective epoch reached their maximum within the epoch. We used a 4 (area) × 2 (offer 1 or 2) ANOVA to test for differences. Neither the main effect of area ($F = 1.96$, $p = 0.297$) nor offer number ($F = 5.04$, $p = 0.110$) was statistically significant (Fig. 3A). Neuronal responses to the onset of offer 1 reached maximum firing (spikes per millisecond) in epoch 1 after an average of 258.55 ms from the offer onset (vmPFC = 265.6 ms; sgACC = 242.5 ms; pgACC = 252 ms; dACC = 274.1 ms). Neuronal responses in epoch 2 reached their maximum, on average, 245.04 ms after the onset of offer 2 (vmPFC = 241.8 ms; sgACC = 242.2 ms; pgACC = 245.6 ms; dACC = 250.5 ms).

We also tested latency using an alternative method. Specifically, we focused on the time at which firing rates reached a significance threshold in response to the variable. We defined latency as the time elapsed from the earliest point in the epoch at which firing rates register as significantly correlated with the expected value until firing rates reached their maximum within the epoch. We used a 4 (area) × 2 (offer 1 or 2) ANOVA to test for differences. Neither the main effect of area ($F = 0.89$, $p = 0.538$) nor offer number ($F = 1.23$, $p = 0.346$) was statistically significant. Using this definition of a significance threshold, we also computed the time elapsed from the onset of the offer to the first time point at which firing rates were significantly correlated with the expected value. We averaged latency across neurons and used a 4 (area) × 2 (offer 1 or 2) ANOVA to test for differences. Neither the main effect of area ($F = 1.14$, $p = 0.457$) nor the offer number ($F = 0.041$, $p = 0.853$) was statistically significant.

Finally, we also calculated latency as the time elapsed from the onset of an offer (in each epoch) to both the peak change in firing rate and to the first significant change in firing rate, irrespective of variable encoding. To do this, we calculated the mean firing rate for each neuron across trials. We then calculated both the mean and standard deviation in firing rate across each epoch. Next, for each time bin, we calculated the absolute value of the difference between firing rate and mean firing rate. We then identified the time bin of the peak change in firing rate relative to the mean firing rate for each neuron. For each epoch, we calculated the time elapsed from the onset of the offer to the peak change in firing rate and averaged these latencies across neurons. We performed a 4 (area) × 2 (offer 1 or 2) ANOVA to look for differences in time elapsed to peak change in firing rate. For both the main effect of area ($F = 0.44$, $p = 0.742$) and the main effect of offer ($F = 3.84$, $p = 0.145$) we found no significant differences. We then defined the time of the first significant change in firing rate as the first time at which firing rates were either greater than 2 standard deviations above or below (to account for neurons with firing rates that decrease) the mean. For each neuron, we then calculated the time elapsed between the onset of the offer and this first significant change in firing rate. For each epoch, we averaged latencies across neurons and performed a 4 (area) × 2 (offer 1 or 2) ANOVA to look for differences in time elapsed to the first significant change in firing rate. For both the main effect of area ($F = 0.76$, $p = 0.448$) and the main effect of offer ($F = 0.17$, $p = 0.912$) we found no significant differences.

## Discussion

Here, we examined neuronal correlates of multiple elements of economic choice in four medial prefrontal cortex regions. Confirming and extending our previous results, we find that these regions show largely similar value-related signals[38,40]. Indeed, by none of the measures we chose did these regions differ qualitatively. This result suggests that the regions do not have conspicuous qualitative differences along the dimensions we studied but leaves open the possibility that they differ quantitatively. Our major finding is that, by several measures, the regions appear to be organized by the functional gradient. First, twelve basic task variables are consistently more decodable later in the gradient. These include both abstract (economic) and spatial variables. Second, the intrinsic timescale is longer later in the gradient. Overall, our results are consistent with the idea that the four regions serve as part of a roughly ventral-to-dorsal functional gradient that gradually transforms neural encodings[10].

The idea that prefrontal regions have a largely gradient-based organization was pioneered by Fuster, who proposed a functional gradient from the sensory to the motor areas and involving the "association cortex" between them[18]. Although he (like many subsequent thinkers) was mainly focused on the lateral prefrontal cortex, the same logic may extend to medial areas. However, the most logical organization of such areas is not obvious, either anatomically or functionally. There are many possibilities. Primarily using anatomical connectivity patterns, Price and colleagues classify all four of our recorded regions in his "medial network", which they propose are responsible for visceromotor functions, and contrasting with the "orbital network", responsible for sensory functions[52,53]. Based on cytoarchitecture and laminar connectivity patterns, Barbas and Pandya[36] take a somewhat different view. For them, areas 25, 24, and 32, as relatively undifferentiated cingulate cortex, are all placed in a similar, low position in a mediodorsal gradient. Area 14, split between the mediodorsal and basoventral trends, occupies a somewhat higher position in the gradient. Our results (although we interpret them differently with respect to a gradient) are not necessarily inconsistent with such a framework, as increased decodability and timescales may simply be a hallmark of less-differentiated PFC regions.

Alternatively, topology would suggest possible ventrodorsal ($14 \rightarrow 25 \rightarrow 32 \rightarrow 24$) or genu-adhering ($25 \rightarrow 14 \rightarrow 32 \rightarrow 24$) gradients. The first one is consistent with the idea that OFC (Price's orbital network) serves as the entryway for economic information to the prefrontal cortex, and area 14 as its next station[3,54]. Our work supports the ventrodorsal hypothesis most strongly, thereby offering the first electrophysiological evidence for one specific medial prefrontal gradient. One prediction of this gradient is that medial area 9 (dorsomedial prefrontal cortex) should be one step above the recorded areas in our analyses[55]. We might also expect that sensory choice information is received by orbitofrontal cortical area 13, and then relayed to the medial prefrontal cortex; thus we would expect area 13 to be below the recorded areas in this gradient, to have shorter intrinsic timescales, and to have less decodable information.

Our results suggest that these four regions have largely overlapping functions in the domain of economic choice. Notably, our results do not imply that these regions have identical functions, nor that their differences are solely quantitative. Indeed, there is plentiful evidence that these regions have important qualitative differences[37,56,57]. To give an example, in a social aggression paradigm, activation of the ventral medial prefrontal cortex correlates with skin conductance response, perhaps reflecting its strong interactions with the hypothalamus and periaqueductal gray, while activation of the dorsal medial prefrontal cortex is more cognitive in nature[58]. Our results do not challenge or invalidate such categorical functional differences. Rather, they suggest that these regions have qualitative differences in some domains and quantitative differences in at least one domain, the domain of economic choice. Indeed, our results do point to a potential limitation too much of traditional functional neuroanatomy. Much of that work is focused exclusively on identifying

the unique contributions of particular regions. While that work is critically important, it necessarily ignores the kinds of brain functions that are not uniquely implemented by specific regions. We believe that economic choice is one such function[10].

We propose that there are essentially three primary organizations the system could take: (1) modular, (2) hierarchical, and (3) a continuum/gradient. In a modular organization, each structure would be responsible for performing some unique and distinct set of computations. The differences between structures would be qualitative. That is, the type of information encoded would differ, but there would not necessarily be differences in the amount of information encoded or the speed at which it is encoded. In a hierarchical organization, information encoding would differ both quantitatively and qualitatively between structures. By contrast, in a gradient organization information encoded by distinct structure would differ primarily only in the amount of encoded information of a given type. There is a general view in neuroeconomics that early reward representations encoded offers and value and later ones encode choice[27]. While there is a large body of empirical work supporting that view, our results are in keeping with several studies, included some of our own, that argue against it. In particular, in previous studies, we have demonstrated robust spatial selectivity in putative early reward regions, an indication of an action plan involving the spatial orientation of the intended choice[43,59]. In keeping with these previous findings, our results support an alternative view. Rather than a goods-to-action transformation, information is the same earlier in the functional gradient as it is later, which is more in line with a cognitive map that tracks choices and potential outcomes. Such representational schemes are particularly likely to be useful in complex and continuous decision-making tasks[60]. One possibility is that there is a modular organization of a somewhat different form than we have proposed. Specifically, it is possible that each region in the series represents a large qualitative step from the one before. Unfortunately, our data are not sufficient to differentiate this organization from smoother ones. Either way, our data are consistent with a series in which each area carries information in a progressively more legible format.

Our results demonstrate a clear monotonic organization of decodability across areas. One important question that our results do not address is the specific form of this monotonic organization. For example, it is possible that the processing occurs in a smooth and gradual form; another possibility is that it occurs in a series of steps; indeed, our data cannot reject the hypothesis that it is steplike for some variables and smooth for others. Successful adjudication between these two hypotheses would require two things, (1) much denser sampling of the medial prefrontal cortex, including medial prefrontal areas that we did not sample at all and sampling across all layers of the cortex, and (2) the development of statistical techniques that can unambiguously dissociate ramping from stepping. We are optimistic that new recording technologies will enable (1) and that the existence of data from (1) will motivate that development of (2). In any case, the specific structure of the organization of change across areas remains an open question.

A broad reading of the electrophysiological literature highlights that many functions have traces that are quite similar in multiple regions[9]. Many scholars draw a distinction between sensory areas, for which a strong modularity case can be made, and "association areas". For example, as Prinz[15] points out, even Fodor, a great advocate of modularity, was more willing to consider distributed function outside of sensory and motor regions[61]. Likewise, Uttal[14] identifies Olds' work on classical (trace) conditioning (1972), which shows that correlates of trace conditioning can be found in nearly every part of the rat brain.

## Methods

**Surgical procedures.** The University Committee on Animal Resources at the University of Rochester and the University of Minnesota approved all animal procedures. Animal procedures were designed and conducted in compliance with the Public Health Service's Guide for the Care and Use of Animals. All of the animals were handled according to approved institutional animal care and use committee (IACUC) protocols (#2005-619 38127A) of the University of Minnesota. The protocol was approved by the Committee on the Ethics of Animal Experiments of the University of Minnesota (NIH permit number: A3456-01). Four male rhesus macaques (Macaca mulatta) served as subjects for both tasks. A small prosthesis head fixation was used. Animals were habituated to laboratory conditions and then trained to perform oculomotor tasks for liquid rewards. We place a Cilux recording chamber (Crist Instruments) over the area of interest (see Behavioral tasks for breakdown). We verified positioning by magnetic resonance imaging with the aid of a Brainsight system (Rogue Research). Animals received appropriate analgesics and antibiotics after all procedures. Throughout both behavioral and physiological recording sessions, we kept the chamber with regular antibiotic washes, and we sealed them with sterile caps.

**Recording sites.** We approached our brain regions through standard recording grids (Crist Instruments) guided by a micromanipulator (NAN Instruments). We recorded neuronal activity from four brain regions: 156 neurons (106 from subject B and 50 from subject H) in vmPFC, 146 neurons (77 from subject B and 69 from subject J) in sgACC area 5, 213 neurons (110 from subject B and 103 from subject V) in pgACC area 32, and 129 neurons (55 from subject B and 74 from subject J) in dACC area 24.

Here, vmPFC (coordinates corresponding to area 14[62]) is defined as the structure rostral to the interaural plane by 29–44 mm, on the coronal plane. On the horizontal plane, it is located from 0 to 9 mm from the brain's ventral surface. On the sagittal plane, the structure is located 0–8 mm from the medial wall (Fig. 2B).

Here, sgACC (coordinates corresponding to area 25[62]) is defined as the structure rostral to the interaural plane by 24–36 mm, on the coronal plane. On the horizontal plane, it is located from 17.33 to 25.12 mm from the brain's dorsal surface. On the sagittal plane, the structure is located 0–5.38 mm from the medial wall (Fig. 2B).

Here, pgACC (coordinates corresponding to area 32[62]) is defined as the structure rostral to the interaural plane by 30.90–40.10 mm, on the coronal plane. On the horizontal plane, it is located from 7.30 to 15.50 mm from the brain's dorsal surface. On the sagittal plane, the structure is located 0–4.5 mm from the medial wall (Fig. 2B).

Here, dACC (coordinates corresponding to area 24[62]) is defined as the structure rostral to the interaural plane by 29.50–34.50 mm, on the coronal plane. On the horizontal plane, it is located from 4.12 to 7.52 mm from the brain's dorsal surface. On the sagittal plane, the structure is located 0–5.24 mm from the medial wall (Fig. 2B).

To confirm the recording sites, we used our Brainsight system. We corroborated the sites against structural magnetic resonance images that were acquired prior to the start of the experiment. These structural images were taken on a Siemens 3 T MAGNETOM Trio Tim, at the Rochester Center for Brain Imagine (0.5 mm voxels). During recording, loci were confirmed by listening for white and gray matter signatures and checked against the Brainsight system, to be within an error of ~1 and ~2 mm in the horizontal and vertical planes, respectively.

For consistency in how we plot our data, each brain area was assigned a color. That color was repeated in all figures corresponding to the data generated from that specific brain area. Importantly, we used an online palette generator specifically designed to allow for selecting color schemes that are visually distinguishable to people with protanopia, deuteranopia, and tritanopia (https://davidmathlogic.com/colorblind).

**Electrophysiological techniques.** Either single (FHC; starting impedance 4 MΩ) or multi-contact electrodes (V-Probe, Plexon) were lowered using a microdrive (NAN Instruments) until waveforms between one and three neuron(s) were isolated. Individual action potentials were isolated on a Plexon system (Plexon, Dallas, TX) or Ripple Neuro (Salt Lake City, UT). Neurons were selected for study solely on the basis of the quality of isolation; we never preselected based on task-related response properties. All cells were hand-sorted using Plexon OLS. All collected neurons for which we managed to obtain at least 300 trials were analyzed; no neurons that surpassed our isolation criteria were excluded from the analysis.

**Eye-tracking and reward delivery.** Eye position was sampled at 1000 Hz by an infrared eye-monitoring camera system (SR Research). Stimuli were controlled by a computer running Matlab (Mathworks) with Psychtoolbox and Eyelink Toolbox. Visual stimuli were colored rectangles on a computer monitor placed 57 cm from the animal and centered on its eyes (Fig. 1A). A standard solenoid valve controlled the duration of juice delivery. Solenoid calibration was performed daily.

**Behavioral tasks.** Four monkeys performed two different tasks with the same basic structure. For the neuronal recordings in vmPFC, subjects B and H performed the risky choice task; and for dACC and sgACC, subjects B and J performed the token

risky choice task (Fig. 2C). Both tasks made use of vertical rectangles indicating reward amount and probability. We have shown in a variety of contexts that this method provides reliable communication of abstract concepts such as reward, probability, delay, and rule to monkeys[63–66].

Risky choice task (vmPFC and pgACC 32). All tasks were based on a standardized general structure for gambling tasks[67–70]. The task presented two offers on each trial. A rectangle 300 pixels tall and 80 pixels wide represented each offer (11.35° of visual angle tall and 4.08° of visual angle wide; Fig. 2B). Two parameters defined gamble offers, reward size, and probability. Two portions divided each gamble rectangle, one red and the other either gray, blue, or green. The size of the color portions signified the probability of winning a small (125 μL), medium (mean 165 μL), or large reward (mean 240 μL), respectively. We drew a uniform distribution between 0 and 100% for these probabilities. Red-colored the rest of the bar; the size of the red portion indicated the probability of no reward. Offer types were selected at random with a 43.75% probability of blue (medium magnitude) gamble, a 43.75% probability of green (high magnitude) gambles, and a 12.5% probability of gray options (safe offers).

On each trial, one offer appeared on the left side of the screen and the other appeared on the right. We randomized the sides of the first and second offers (left and right). Each offer appeared for 400 ms and was followed by a 600-ms blank period. After the offers were presented separately, a central fixation spot appeared, and the monkey fixated on it for 100 ms. Following this, both offers appeared simultaneously and the animal indicated its choice by shifting gaze to its preferred offer and maintaining fixation on it for 200 ms. Failure to maintain gaze for 200 ms did not lead to the end of the trial but instead returned the monkey to a choice state; thus monkeys were free to change their mind if they did so within 200 ms (although in our observations, they seldom did so). Following a successful 200-ms fixation, the trial immediately resolved the gamble and delivered the reward. We considered trials that took ~7 s as inattentive trials and we did not include them in the analyses (this removed ~1% of trials). Outcomes that yielded rewards were accompanied by a visual cue: a white circle in the center of the chosen offer. All trials were followed by an 800-ms intertrial interval with a blank screen.

Token risky choice task (sgACC 25 and dACC 24). Another similarly structured gambling task, where gambles each had two potential outcomes, wins or losses in terms of "tokens" displayed on the screen as cyan circles. A small reward (100 μL) was administered concurrently with gamble feedback on each trial, regardless of gamble outcome. Trials in which the monkey accumulated six or more tokens triggered an extra "jackpot" epoch in which a very large reward (300 μL) was administered (Fig. 2C).

**Behavioral analysis**. To confirm the statistical validity of the behavioral results, we first identified the chosen offer (first or second) for each trial. We then calculated the proportion of trials, for each recording session, in which the offer with the higher value was chosen. We then analyzed the vector of choice proportions using a one-sample *t*-test, to determine if the average proportion of greater value choices was statistically different from zero.

**Reuse of data**. Some of these data were previously published (vmPFC dataset in Strait et al.[38]; sgACC and dACC data sets in Azab and Hayden, 2017[39]; data from pgACC have not been previously published).

**Statistical methods**. We constructed peristimulus time histograms by aligning spike rasters to the presentation of the first offer and averaging firing rates across multiple trials. We calculated firing rates in 20-ms bins, but we generally analyzed them in longer (500 ms) epochs. For display, we smoothed peristimulus time histograms using a 200-ms running boxcar. Some statistical tests of neuron activity were only appropriate when applied to single neurons because of variations in response properties across the population. In such cases, a binomial test was used to determine if a significant portion of single neurons reached significance on their own, thereby allowing conclusions about the neural population as a whole.

**Offer encoding latency**. We computed an average latency score for each area and both offers. First, we isolated firing rates for both epochs, as each was binned to the onset of one of the offers. Each epoch consisted of a 500 ms window constituted by 20 ms bins. Next, we calculated the average firing rate for each neuron, for each 20 ms bin, across trials. Then, we determined how much time (in ms) passed for a given neuron to reach its peak firing rate for the epoch. Finally, we calculated the average latency to peak firing rate across all neurons in the region.

**Intrinsic timescales**. To measure intrinsic timescales, we followed similar steps previously described[32]. We isolated a 2-s time window preceding the onset of offer 1, to remain independent of trial variables. Using a 20 ms sliding window, we then computed the autocorrelation for the 2 s window with a given lag $k\Delta$ between time $i$ and time $j$, where $k\Delta = |i − j|$. The lag ranged from 20 and 720 ms. We then determined that the autocorrelation decay in each structure could be well-fit by an exponential decay function

$$R(k\Delta) = A[\exp(−k\Delta/\tau) + B],$$

where $A$ = the amplitude of the autocorrelation, $k\Delta$ = the lag, $\tau$ = intrinsic time-scale, and $B$ = the offset to account for long timeframes outside of the measured window. This formula follows what was previously described[32].

**Decoding analysis**. We built a pseudo-population of pseudo-trials. First, we isolated each epoch and collapsed the firing rates for each trial into an average for the 500 ms period. Then, we separated the data set for each neuron by the given variable label (choice of first or second offer; choice of left or right offer; offer one position left or right; offer two positions left or right; offer one value higher or lower than the mean value and offer value higher or lower than the mean). We randomly selected 1000 samples for each neuron resulting in $2n \times 1000$ matrices (one for each label level), where n represented the number of neurons recorded from each region. This constituted the pseudo-population or pseudo-trials. To execute the decoder, each matrix was split in half and concatenated with the half from the other label. We used one of these matrices to train a binary support vector machine, the other was used for cross-validation. We used the trained model to predict the binary label for each pseudo-trial in the cross-validation set. We then compared the predicted outcome to the known choice outcome and an accuracy rate was calculated across pseudo-trials. This process was repeated 1000 times for each target structure and for each of the given labels and epochs to get a distribution of accuracy rates. Thus, the standard error of the mean, used in displaying the error bars, represents the standard error over the variance of the cross-validations. In addition, the exact process was repeated on randomly shuffled data, to confirm that expected prediction accuracy was 50% when randomized.

We performed a Spearman correlation across decoding accuracy distributions from all structures. For each decoder, we generate a distribution of decoder accuracies from 1000 permutations per brain area. Each accuracy in the distribution was assigned a value from 1 to 4, corresponding to the proposed gradient order (1 = vmPFC, 2 = sgACC, 3 = pgACC, and 4 = dACC). We then aggregated the distributions into a single matrix and performed a Spearman correlation between the distribution of accuracies and the assigned gradient order.

**Reporting summary**. Further information on research design is available in the Nature Research Reporting Summary linked to this article.

## Data availability

The data that support the findings of this study are available from the corresponding author upon reasonable request. Source Data, the relevant raw data used to generate each figure, is available as a Source Data file on Dryad (https://doi.org/10.5061/dryad.18931zcxv) and provided with this manuscript. Source data are provided with this paper.

## Code availability

The original code used to analyze the data in this study is available from the corresponding author upon reasonable request.

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

## Acknowledgements

We thank Meghan Castagno Pesce, Marc Mancarella, Caleb Strait, and Tommy Blanchard for assistance with data collection, and the rest of the Hayden/Zimmermann lab for valuable discussions. This research was supported by a National Institute on Health grant R01 DA038106 (to B.Y.H.) an R01 MH118257 (to S.R.H.), a National Institute on Drug Abuse grant P30 DA048742-01A1 (to B.Y.H., S.H., and J.Z.), a National Institute for Biomedical Imaging Grant P41 EB027061 (to B.Y.H. and J.Z.), and a UMN AIRP award (to B.Y.H., S.H., and J.Z.).

## Author contributions

D.M. was responsible for the conception of the study, the analyses, and the preparation of the paper. T.C. and M.W. contributed to the collection of the behavioral and neuro-physiological data. S.H. contributed substantially to the neuroanatomical framework and writing at the backbone of the observed gradient. B.H. and J.Z. contributed funding, resources, and lab space, and were responsible for the conceptualization of the study and the preparation of the paper.

## Competing interests

The authors declare no competing interests.
