## [Peer Review File · Nature Communications]

REVIEWER COMMENTS

Reviewer #1 (Remarks to the Author):

This study by Maisson and colleagues compares the functional properties of neurons in four medial prefrontal cortical regions in macaques making economic decisions. The objective was to determine whether a hierarchical relationship between these regions could be inferred on the basis of the encoding of task-relevant variables. In addition to new data, the study is based upon data reported in prior studies; however, the analyses are novel. The two economic choice tasks used in the study are similar: both require that the monkeys choose between stimuli that differ according to the magnitude and probability of a juice reward. In general, the monkeys' decisions reflect the expected values (product of magnitude and probability) of the offers, indicating that the subjects understand the task. Neural data were obtained from single units within four predefined ROIs within the medial frontal cortex: ventromedial PFC, subgenual ACC, pre-genual ACC, dorsal ACC, corresponding to cytoarchitecturally defined areas 14, 25, 32, and 24, respectively. A hierarchical relationship among these regions (14 -> 25 -> 32 -> 24) was identified on the basis of two main metrics: a decay function fit to the firing rate autocorrelations measured at different temporal lags; and the accuracy of a neural classifier for several relevant variables. The authors conclude that these data indicate the existence of a 'roughly ventral-to-dorsal functional gradient' within the primate medial frontal cortex.

The core question of this study is a pertinent one: while this part of the frontal lobe has long been associated with reward and decision-making, there is still no clear understanding of the distinct roles of its many sub-regions. The results therefore have the potential to make a tangible positive impact upon the field. The experiments and analyses are all technically sound, and so the basic results as they are presented are not in doubt.

These strengths aside, I am not fully convinced that the data support the conclusion that these four regions constitute an actual hierarchy, as opposed to a functional gradient.

The first reason is that the conclusion rests upon only two observations: the autocorrelation time constant, and the classifier performance. In contrast, measurements of encoding latency did not indicate a hierarchy or directionality of information processing (though see note 5 below); nor did the three 'signatures of the choice process' shown in Figure 5. I am curious about whether there are other measures that might give additional support to the proposed hierarchy. Examples include basic measures such as baseline firing rates or measurements of firing rate variability (e.g. fano factor), or more nuanced metrics such as noise correlations (if available), or intrinsic dimensionality. Likewise, I am curious about the encoding/decodability of other task variables identified in several prior studies, such as the encoding of the chosen/unchosen object values, or of choices and rewards on the previous trial. In other words, I would urge the authors to consider a few other key metrics and task variables (such as the ones mentioned above) that could be used to build a more complete/convincing body of evidence for the organization among these structures.

The second reason that I am not convinced that the data point to a hierarchical relationship is that the two metrics do not obviously suggest a transformation or 'untangling' of choice-related information. The classic example of a hierarchy is of course the primate visual system, which begins with the encoding of center-surround receptive fields, and builds gradually to high-level features such as motion, objects identity, etc. It seems reasonable that this kind of transformation is what one would look for when identifying hierarchical relationships.

While the results certainly do show an overall functional gradient (e.g. from low to high decoder performance, Figure 7), they do not appear to show any sort change in the /kind/ of information encoded. For example, if vmPFC activity had showed more accurate decoding of value-related variables, whereas dACC had more accurate decoding of space- or action-related variables, that would suggest as a transformation of task-related information from a goods-based to action-based reference frame, which would be consistent with a hierarchical relationship. As another example, if the degree of 'feature integration' (Fig 5, row A) was low in vmPFC, but high in dACC, this would again suggest a transformation of the relevant variables.

In summary, the data as presented here certainly indicate a functional gradient (a change in quantity but not quality) with respect to two of the measures that were considered. I do not think that this is enough to conclude that true functional hierarchy exists. I realize that the distinction between a hierarchy and a gradient may seem like splitting hairs, but the paper's introduction explicitly endorses the idea that a hierarchy involves transformation or 'untangling' of information (lines 69-80), and so that is the criterion by which the data should be judged.

In addition to the concerns with interpretation above, there are additional suggestions and comments, mostly technical in nature:

1) I am curious about the minor discrepancy between the encoding results (Figure 4) and the decoding results (Figure 7), specifically the fact that the vmPFC shows the lowest decoding of ev1-epoch1 in Fig. 7, but has the 2nd highest proportion of neurons encoding this variable in Fig 4. Is this due to some intrinsic property of vmPFC neural activity, e.g. low average firing rates leading to noisier decoding? I would like to have some more intuition (or data) to explain this discrepancy.

2) Related to the above, I am curious whether the differential performance of the classifier between regions could be in part due to differences in the number of neurons used for each area. (These numbers were, frustratingly, not provided). What would the results look like if the same number of neurons were used for all areas -- i.e. if each area were matched with respect to the number of features available to the SVM? How quickly does decoder performance degrade for each region as neurons are dropped from the decoder one-by-one?

3) A final concern with the classifier is that the classification of expected values (a continuous variable) into high/low (a binary variable) obscures potentially informative information about how the regions encode information. That is, the classifier would not distinguish between a region with binary-like firing responses for values above/below the mean, and a region with graded firing responses that scale monotonically with expected value. (This distinction in particular is relevant for the identification of a functional hierarchy, which is characterized by the transformation of information, as discussed above.) As I recall, Rich and Wallis (2016) used a multinomial classifier for value stimuli that varied from 1-4 reward units. I think a similar approach might be warranted here, e.g. to divide expected value into 4 or more bins, and to determine whether individual regions support graded vs. binary-like decoding of expected value.

4) The latency analysis is unsatisfying, because it considers only the response peaks, and not the response onset, i.e. the time that a given response first exceeds chance levels. Both of these are potentially useful in identifying a hierarchical relationship.

5) As I understand it, the intrinsic time-scale analysis uses the 2s of firing data preceding the onset of the first target. I'd like to know more about what exactly is happening in that 2s time span. Is there an initial fixation period? If so, how long is it? The high level concern here is that the 2s window used for this analysis is free of any major task events that might influence the time-scale calculations.

6) Lines 566-568, describing the decoding analysis, are a little confusing. For example does "1st offer and second" mean "whether the 1st or 2nd offer was chosen"? A little clarity here would be helpful.

7) What procedures were used for spike sorting?

8) A few little graphical suggestions: In Figure 3B-E, it's unclear why so much data is shown after ~2s; I would suggest omitting these data, so that the more informative section of the graph (~0-2.5s) is relatively larger and easier to parse. In Figure 4A, I would recommend adding shading or dotted lines to indicate the major trial epochs, as in Figure 3B-E. In Figure 5, both the X and Y axes could be 'zoomed in' to give a more informative view of the data. In Figure 7, the alphabetical labels (A-H) could be shifted to the right, to better align with the center of each bar cluster.

Reviewer #2 (Remarks to the Author):

In their manuscript 'A functional hierarchy for choice in medial prefrontal cortex' Maisson and co-authors report a re-analysis of electrophysiological recordings from four mPFC regions in rhesus monkeys performing value-based reasoning (gambling) tasks. The authors found that all regions encode economic variables. However, they observed increasing decodability, i.e. coding strength, and reducing decay in the spike-count autocorrelation functions, a potential signature of increasing integration capacity ('intrinsic timescales'), along a ventro-dorsal axis. Maisson and co-workers interpret these findings as evidence for a hierarchical organization of primate mPFC in economic choices.

Overall, I think this is a solid study. The topic and the experimental and analytical approaches, although not overly original or novel, should be of interest to a wider audience. The paper is comparatively brief, concise, and to-the-point. The results are presented in a clearly understandable manner for the most part. I only have a few comments and relatively minor suggestions for a revised version:

- The term 'hierarchy', frequently used throughout the manuscript, implies a sequence, an ordered set of operations. Given that response latencies are identical in the investigated areas and that the authors do not present functional connectivity analyses I think there is little evidence to support this rather strong term. I would suggest to use 'gradient' instead (as already done in a few instances), which is more neutral.
- Brain area numbering should be included in Fig. 2B; consider showing this figure earlier since it is necessary for the non-experts to understand the neuroanatomical backdrop of the study
- Figure 3B-E should be complemented by a population response (including a mentioning of the total number of neurons per panel)
- The present analysis of differing selectivity to various economic variables (Fig. 4) is a little contorted and could be improved. I would suggest to replace the 'proportion of neurons with selective firing rates' with the measure of explained variance. The EV can be plotted in a time-resolved manner for all variables of interest and for all brain regions. I think this would allow the reader to more quickly pick up the role of the analysed brain regions in coding particular elements of value-based reasoning.
- The layout and graphical design of the panels with time-resolved activity in Fig. 3 and Fig. 4 should be the same (i.e. choice of dashed or solid lines, marked task epochs etc)
- l. 304: should this read eight variables instead of seven?

Reviewer #1 (Remarks to the Author):

The core question of this study is a pertinent one: while this part of the frontal lobe has long been associated with reward and decision-making, there is still no clear understanding of the distinct roles of its many sub-regions. The results therefore have the potential to make a tangible positive impact upon the field. The experiments and analyses are all technically sound, and so the basic results as they are presented are not in doubt.

Reviewer 1, Comment 1:

These strengths aside, I am not fully convinced that the data support the conclusion that these four regions constitute an actual hierarchy, as opposed to a functional gradient.

The first reason is that the conclusion rests upon only two observations: the autocorrelation time constant, and the classifier performance. In contrast, measurements of encoding latency did not indicate a hierarchy or directionality of information processing (though see note 5 below); nor did the three 'signatures of the choice process' shown in Figure 5.

Reviewer 1, Response 1:

The reviewer points to an important distinction, that between (to use the reviewer's terms) a *hierarchy* and a *gradient*. We agree with the reviewer's interpretation of our data, and also that our original terms were potentially misleading. We have made the following two changes (1) we now use the reviewer's recommended term 'gradient' instead of 'hierarchy' throughout the manuscript (including in the title), and (2) we now provide additional text explaining what we mean by these terms.

In our explanation, we identify three possible organizations (and we suspect the reviewer would agree): (1) modular, (2) hierarchical, and (3) gradient (or continuum). The major result of our paper is that it argues against the modular organization and in favor of the other two, without strongly distinguishing between them. In the reviewer's view, our claim is more specific. If our understanding is correct, then our terms were misleading, and so changing to gradient (which refers to 3) is more appropriate.

The revised title is:

A functional gradient for choice in the medial prefrontal cortex.

Revised Abstract:

We have previously proposed that choice-relevant brain regions have overlapping functions and can be organized into a series that progressively transforms information about options into choices. Here, we examined responses of neurons in four regions of the medial prefrontal cortex as macaques performed two-option risky choices. All four regions encoded economic variables in similar proportions and showed similar putative signatures of key choice-related computations. We found evidence for the predicted gradient of function that proceeds from areas 14→25→32→24. Specifically, we found that decodability of eleven distinct task variables increased along that path, consistent with the idea that anatomically later regions make choice-relevant variables more separable. We also found longer intrinsic timescales in the same series. Together these results highlight the importance of the medial wall in choice, endorse a specific

gradient-based organization, and argue against a modular functional neuroanatomy of choice.

New paragraph added to the discussion:

We propose that there are essentially three primary organizations the system could take: (1) modular, (2) hierarchical, and (3) a continuum/gradient. In a modular organization, each structure would be responsible for performing some unique and distinct set of computations. The differences between structures would be qualitative. That is, the type of information encoded would differ, but there would not necessarily be differences in the amount of information encoded or the speed at which it is encoded. In a hierarchical organization, information encoding would differ both quantitatively and qualitatively between structures. By contrast, in a gradient organization information encoded by distinct structure would differ primarily only in the amount of encoded information of a given type. There is a general view in neuroeconomics that early reward representations encode offers and value and later ones encode choice (Cai and Padoa-Schioppa, 2014). While there is a large body of empirical work supporting that view, our results are in keeping with several studies, included some of our own, that argue against it. In particular, in previous studies we have demonstrated robust spatial selectivity in putative early reward regions, an indication of an action plan involving the spatial orientation of the intended choice. In keeping with these previous findings, our results support an alternative view. Rather than a goods-to-action transformation, information is the same earlier in the functional gradient as it is later, which is more in-line with a cognitive map that tracks choices and potential outcomes.

Reviewer 1, Comment 2:

I am curious about whether there are other measures that might give additional support to the proposed hierarchy. Examples include: basic measures such as baseline firing rates or measurements of firing rate variability (e.g. fano factor)

Reviewer 1, Response 2:

We thank the reviewer for these excellent and useful suggestions that further probe whether the data are best described by a strict hierarchy or a functional gradient. Briefly, none of these measures provide clear guidance on the question of hierarchy, which, in our view, speaks in favor to the gradient versus hierarchy position.

To measure baseline firing rates in each structure, we computed the average firing rate for each neuron during the 1-second pre-trial period, from -2 s to 0 s, where time 0 was set to the onset of offer 1. (Note that this is the same time window on which we performed our analysis of intrinsic timescales. The baseline firing rate in vmPFC was 3.11 spikes/second. The baseline firing rate in sgACC was 3.69 spikes/second. The baseline firing rate in pgACC was 0.75 spikes/second. Finally, the baseline firing rate in dACC was 5.06 spikes/second. A Spearman correlation of these baseline firing rates with the proposed order was not significant ($r = 0.4$, $p = 0.75$, Spearman correlation). As the results are non-significant, we cannot draw any additional conclusions from this analysis.

To measure firing rate variability, we computed the average Fano Factor (FF) across neurons in each region, replicating the procedure outlined in Chang et al. (2012). For each time bin (20 ms) in each epoch (500 ms), we computed the average firing rate (FR)

from across trials. We then calculated the variance and mean for firing rates across all time bins, within the given epoch. For each cell, the FF was calculated as the ratio of firing rate variance to average firing rate across the epoch. We then calculated the average FF across cells. As a result, in epoch 1: vmPFC FF = -0.14, sgACC FF = 2.08, pgACC FF = 0.29, dACC FF = -1.96; epoch 2: vmPFC FF = -0.17, sgACC FF = 0.05, pgACC FF = 0.09, dACC FF = 0.22. We computed a Spearman correlation between the FF and the proposed hierarchical order, for each epoch. In epoch 1, the correlation was not significant ($r = -0.4$, $p = 0.75$). In epoch 2, the correlation was again not significant ($r = 1$, $p = 0.083$). As the results are non-significant, we cannot draw any additional conclusions from this analysis.

Reviewer 1, Comment 3:

[the authors could also try a] more nuanced metrics such as noise correlations (if available), or intrinsic dimensionality.

Reviewer 1, Response 3:

We thank the reviewer for making this insightful suggestion. To measure intrinsic dimensionality, we performed a principal component analysis on the normalized firing rates across neurons in each of the offer epochs, independently for each structure. For each neuron, we isolated the firing rates from both offer epochs. We calculated the mean firing rate across trials for each 20 ms time bin. Using these average responses, we composed a matrix of time X neurons. We performed an eigenvalue decomposition and computed the explained variance due to each of the first three principal components (PCs). We then calculated the degree of change, by calculating the slope, in explained variance from the first PC to the second PC, and again from the second PC to the third. We performed this same analysis for each offer epoch independently. Finally, we asked if the change in explained variance between the first three PCs was significantly correlated with the proposed order. We found that in epoch 1, the change in explained variance between PC1 and PC2 did not significantly change with gradient order ($\rho = -1$, $p = 0.083$, Spearman correlation). Similarly, the change in explained variance between PC2 and PC3, in epoch 1, did not significantly change with gradient order ($\rho = -1$, $p = 0.083$, Spearman correlation). We found that in epoch 2, the change in explained variance between PC1 and PC2 did not significantly change with gradient order ($\rho = -1$, $p = 0.083$, Spearman correlation). The change in explained variance between PC2 and PC3, in epoch 2, did not significantly decrease with gradient order ($\rho = -0.4$, $p = 0.75$, Spearman correlation). As the results are not significant, we cannot draw any additional conclusions from this analysis.

Reviewer 1, Comment 4:

Likewise, I am curious about the encoding/decodability of other task variables identified in several prior studies. (In other words, I would urge the authors to consider a few other key metrics and task variables (such as the ones mentioned above) that could be used to build a more complete/convincing body of evidence for the organization among these structures.)

- such as the encoding of the chosen/unchosen object values,
- or of choices and rewards on the previous trial.

Reviewer 1, Response 4:

The reviewer rightly suggests adding additional independent variables to our analysis, and mentions four: (1) chosen values, (2) unchosen values, (3) choice on previous trial, (4) reward on previous trial.

We added these variables to the classifier analysis. In brief, the new analyses confirm and extend the claims we made in our original paper:

Next, we looked at how accurately the expected value of the chosen offer could be decoded from firing rates in the choice epoch (**Figure 6F**). The classifier decoded chosen value on each trial (binomial test, $p < 0.0001$) for all regions: vmPFC (66.1%), sgACC (69.1%), pgACC (69.8%), and dACC (70.8%). When we compared the proposed gradient order with the decoding accuracy distributions, decoding accuracy increased with gradient order ($\rho = 0.59$, $p < 0.001$, Spearman's correlation).

Next, we looked at how accurately the expected value of the unchosen offer (offer 1 or 2) could be decoded from firing rates in the choice epoch (**Figure 6G**). The classifier decoded choice on each trial (binomial test, $p < 0.0001$) for all regions: vmPFC (67.1%), sgACC (68.1%), pgACC (69.3%), and dACC (66.4%). When we compared the proposed gradient order with the decoding accuracy distributions, decoding accuracy did not significantly increase with gradient order ($\rho = -0.03$, $p = 0.058$, Spearman's correlation).

Next, we looked at how accurately the chosen offer (offer 1 or 2) from the previous trial could be decoded from firing rates in the current choice epoch (**Figure 6H**). The classifier decoded choice on each trial (binomial test, $p < 0.0001$) for all regions: vmPFC (61.3%), sgACC (67.1%), pgACC (67.4%), and dACC (67.9%). When we compared the proposed gradient order with the decoding accuracy distributions, decoding accuracy increased with gradient order ($\rho = 0.51$, $p < 0.001$, Spearman's correlation).

Next, we looked at how accurately the experienced reward (rewarded or not rewarded) from the previous trial could be decoded from firing rates in the current choice epoch (**Figure 6I**). The classifier decoded choice on each trial (binomial test, $p < 0.0001$) for all regions: vmPFC (66.7%), sgACC (68.1%), pgACC (72.1%), and dACC (70.3%). When we compared the proposed gradient order with the decoding accuracy distributions, decoding accuracy increased with gradient order ($\rho = 0.66$, $p < 0.001$, Spearman's correlation).

Reviewer 1, Comment 5:

The second reason that I am not convinced that the data point to a hierarchical relationship is that the two metrics do not obviously suggest a transformation or ‘untangling’ of choice-related information. The classic example of a hierarchy is of course the primate visual system, which begins with the encoding of center-surround receptive fields, and builds gradually to high-level features such as motion, objects identity, etc. It seems reasonable that this kind of transformation is what one would look for when identifying hierarchical relationships.

Reviewer 1, Response 5:

We thank the reviewer for this valuable insight. As noted above, we think this conceptual issue is largely resolved by a change in terminology from 'hierarchy' to 'gradient'.

We would also note a critical issue: we don't actually know what information neurons in the PFC are in the business of representing - all we can do is see how those representations project onto the variables we happen to choose. Because those projections are not isomorphic to the underlying representations, we would almost inevitably expect quantitative, not qualitative, differences between areas, even if the core organizational principle is one of untangling. We now make this brief point clearly in the revised Discussion.

Reviewer 1, Comment 6:

While the results certainly do show an overall functional gradient (e.g. from low to high decoder performance, Figure 7), they do not appear to show any sort change in the "kind" of information encoded.

Reviewer 1, Response 6:

The reviewer is entirely correct and, indeed, this is a key finding. We apologize for not making this point clearer. We hope the revisions do so.

We have add this paragraph to the Discussion section:

We propose that there are essentially three primary organizations the system could take: (1) modular, (2) hierarchical, and (3) a continuum/gradient. In a modular organization, each structure would be responsible for performing some unique and distinct set of computations. The differences between structures would be qualitative. That is, the type of information encoded would differ, but there would not necessarily be differences in the amount of information encoded or the speed at which it is encoded. In a hierarchical organization, information encoding would differ both quantitatively and qualitative between structures. By contrast, in a gradient organization information encoded by distinct structure would differ primarily only in the amount of encoded information of a given type. There is a general view in neuroeconomics that early reward representations encoded offers and value and later ones encode choice (Cai and Padoa-Schioppa, 2014). While there is a large body of empirical work supporting that view, our results are in keeping with several studies, included some of our own, that argue against it. In particular, in previous studies we have demonstrated robust spatial selectivity in putative early reward regions, an indication of an action plan involving the spatial orientation of the intended choice. In keeping with these previous findings, our results support an alternative view. Rather than a goods-to-action transformation, information is the same earlier in the functional gradient as it is later, which is more in-line with a cognitive map that tracks choices and potential outcomes.

Reviewer 1, Comment 7:

For example, if vmPFC activity had showed more accurate decoding of value-related variables, whereas dACC had more accurate decoding of space- or action-related variables, that would suggest as a transformation of task-related information from a goods-based to action-based reference frame, which would be consistent with a hierarchical relationship.

As another example, if the degree of 'feature integration' (Fig 5, row A) was low in vmPFC, but high in dACC, this would again suggest a transformation of the relevant variables.

Reviewer 1, Response 7:

We agree that some implementations of a hierarchical organization would predict a change from goods to action space. As the reviewer rightly points out, we obviously don't see that.

There is a general view in the field that early reward representations are likely to be "goods based" and later ones are "action based." There is, of course, a large body of empirical work supporting that view. Our data presented here do not support that view. Indeed, several of our past studies argue against it as well; in particular, we have demonstrated robust spatial selectivity in early reward regions (Strait et al., 2016; Yoo et al., 2018; Mehta et al., 2019). This is not to say our work is particularly innovative or outstanding; many other groups have reported similar findings.

Instead, what we see here is that the information is the same in early and late regions, it is just more decodable later. We think that finding is important, and tells us something critical about what's going on along the medial wall of the PFC.

We think that our results provide suggestive evidence that the transformation from ventral to dorsal regions is not a goods to action transform, but instead is one that involves greater decodability - that is, more linearly separable - in other words, untangling. We think that this finding is important and now highlight it in a new paragraph in the Discussion:

There is a general view in neuroeconomics that early reward representations encoded offers and value and later ones encode choice (Cai and Padoa-Schioppa, 2014). While there is a large body of empirical work supporting that view, our results are in keeping with several studies, included some of our own, that argue against it. In particular, in previous studies we have demonstrated robust spatial selectivity in putative early reward regions, an indication of an action plan involving the spatial orientation of the intended choice. In keeping with these previous findings, our results support an alternative view. Rather than a goods-to-action transformation, information is the same earlier in the functional gradient as it is later, which is more in-line with a cognitive map that tracks choices and potential outcomes.

Reviewer 1, Comment 8:

In addition to the concerns with interpretation above, there are additional suggestions and comments, mostly technical in nature:

I am curious about the minor discrepancy between the encoding results (Figure 4) and the decoding results (Figure 7), specifically the fact that the vmPFC shows the lowest decoding of ev1-epoch1 in Fig. 7, but has the 2nd highest proportion of neurons encoding this variable in Fig 4. Is this due to some intrinsic property of vmPFC neural activity, e.g. low average firing rates leading to noisier decoding? I would like to have some more intuition (or data) to explain this discrepancy.

Reviewer 1, Response 8:

The reviewer raises a really interesting point. On the one hand, the proportion of recorded neurons with firing rates correlated to offer value does not appear to track

proposed gradient order. On the other hand, the accuracy with which offer value information can be decoded from neural responses does increase with gradient order. That is, while the amount of neurons encoding offer value in each structure does not increase with gradient order, decodability does. The sheer proportion of neurons showing modulation by a given variable, in a given structure, does not reflect the reliability with which downstream decoders can interpret the output. For example, one structure may have a large amount of neurons that each only encode a tiny, but statistically reliable, amount of offer value information. Another structure may have very few neurons that encode value, but the extent to which they do is large, allowing a downstream structure to more easily decode the output. Indeed, in a recent publication (Tang et al., 2021), the authors found similar results in the lateral PFC, and drew the same conclusion; that fewer cells encoding information more accurately resulted in better decoding accuracy than simply having more significantly tuned cells. Additionally, SVM decoders are sensitive to the variance structures along each dimension of the underlying matrix, which includes all neurons and all trials simultaneously. They also take into consideration any higher order linear transformations. By contrast, correlating average firing rate within a given neuron with a given variable, like value, does not account for population-level variance structures. Thus, these two analyses offer distinct and complementary interpretation. The first allows us to identify how many individual neurons are likely to participate in encoding of value. The second allows us to determine how the population as a whole carries decodable information about value and choice. The reviewer raises an important point and we hope that future studies with simultaneously recorded neurons can help us elucidate this further.

Reviewer 1, Comment 9:

Related to the above, I am curious whether the differential performance of the classifier between regions could be in part due to differences in the number of neurons used for each area. (These numbers were, frustratingly, not provided). What would the results look like if the same number of neurons were used for all areas -- i.e. if each area were matched with respect to the number of features available to the SVM? How quickly does decoder performance degrade for each region as neurons are dropped from the decoder one-by-one?

Reviewer 1, Response 9b:

To address the reviewer's important concern, we blindly reduced the size of our datasets until each had the same number of cells ($n = 125$). Doing this necessarily introduces variability and noise (non-systematic error) without introducing systematic error. Despite this, we observed functionally identical results. Specifically, for all of our twelve measures the ordering was unchanged (and as a consequence, the Spearman result was unchanged). Overall, these results robustly confirm the initial findings we presented in our manuscript and demonstrate they are not an artifact of differential set sizes. We have added these clarifications to the manuscript:

We next asked whether the number of neurons collected in each region had some effect on measures of decodability. We were concerned that the different number of neurons collected for each region affected decodability. Specifically, we decimated our datasets until each had the same number of cells (specifically, 125, a number chosen before analysis). We observed that for all of our twelve measures the order was unchanged. Importantly, when we averaged the decoding accuracy across all decoders, we found a significant increase with gradient order. We conclude from this control

analysis that our original findings are robust and do not depend on the number of cells in each sample.

Reviewer 1, Response 9a:

We apologize for not including information about the number of neurons. The information has been added to both the results and methods sections:

We recorded neuronal activity from four brain regions: 156 neurons (106 from subject B and 50 from subject H) in ventromedial prefrontal cortex (vmPFC), 146 neurons (77 from subject B and 69 from subject J) in subgenual anterior cingulate area 5 (sgACC), 213 neurons (110 from subject B and 103 from subject V) in pregenual anterior cingulate area 32 (pgACC), and 129 neurons (55 from subject B and 74 from subject J) in dorsal anterior cingulate area 24 (dACC).

Reviewer 1, Comment 10:

A final concern with the classifier is that the classification of expected values (a continuous variable) into high/low (a binary variable) obscures potentially informative information about how the regions encode information. That is, the classifier would not distinguish between a region with binary-like firing responses for values above/below the mean, and a region with graded firing responses that scale monotonically with expected value. (This distinction in particular is relevant for the identification of a functional hierarchy, which is characterized by the transformation of information, as discussed above.) As I recall, Rich and Wallis (2016) used a multinomial classifier for value stimuli that varied from 1-4 reward units. I think a similar approach might be warranted here, e.g. to divide expected value into 4 or more bins, and to determine whether individual regions support graded vs. binary-like decoding of expected value.

Reviewer 1, Response 10:

To address this concern, we added a regression-based multinomial classifier for decoding multiple EV bins. Unlike Rich and Wallis (2016), our expected value variable is continuous. Therefore, we discretized expected value into bins in order to perform the multinomial classifier. To do this, while still ensuring there would be a usable number of trials in each bin, we separate expected value into 6 equally-sized, consecutive bins. We separated trials based on the corresponding binned EVs and used these separated responses to construct pseudo-samples, following the same approach already described in our manuscript. We then trained the SVM, and used the model to predict the offer and the chosen EV bins on a novel cross-validation set. We found that the proposed gradient in the majority of our other findings (vmPFC->sgACC->pgACC->dACC) was also evident when we decoded these variables using multiple bins. As the results remain unchanged from the previous report, we included these additional confirmatory results as a reported control procedure. We have added this text to the Results section:

Another possible confound comes from the fact that expected value here is a continuous variable. If some structures have graded coding of the variable, a binary classifier may not be sensitive to this property. To address this concern, as an additional control, we performed a regression-based multinomial classifier for decoding multiple EV bins. To discretize EV, while still ensuring there would be a usable number of trials in each bin, we separated expected value into 6 equally-sized, consecutive bins. We

separated trials based on the corresponding binned EVs and used these separated responses to construct pseudo-samples, following the same approach already described. We then trained the SVM, and used the model to predict the offered and the chosen EV bins on a novel cross-validation set. We found that the proposed gradient in the majority of our other findings (vmPFC->sgACC->pgACC->dACC) was also evident when we decoded these variables using multiple bins. As the results remain unchanged from the binary decoding of EV, we do not discuss them further.

Reviewer 1, Comment 11:

The latency analysis is unsatisfying, because it considers only the response peaks, and not the response onset, i.e. the time that a given response first exceeds chance levels. Both of these are potentially useful in identifying a hierarchical relationship.

Reviewer 1, Response 11:

We have added the recommended analysis. For each neuron, we now compute the latency as the time elapsed from when its firing rate first registers as significantly correlated with an offer's expected value until the time to peak firing rate. (For this modified analysis, we excluded those neurons that never showed a significant correlation.) That is, latency now indicates the average time lapsed, across neurons, from when firing rates surpassed significance until they reach peaked response. The statistical significance of these results confirms what was originally reported. We have modified the corresponding text:

We also tested latency using an alternative method. Specifically we focused on the time at which firing rates reached a significance threshold in response to the variable. We defined latency as the time elapsed from the earliest point in the epoch at which firing rates register as significantly correlated with expected value until firing rates reached their maximum within the epoch. We used a 4 (area) x 2 (offer 1 or 2) ANOVA to test for differences. Neither the main effect of area ($F = 0.89$, $p = 0.538$) nor offer number ($F = 1.23$, $p = 0.346$) was statistically significant.

Reviewer 1, Comment 12:

As I understand it, the intrinsic time-scale analysis uses the 2s of firing data preceding the onset of the first target. I'd like to know more about what exactly is happening in that 2s time span. Is there an initial fixation period? If so, how long is it? The high level concern here is that the 2s window used for this analysis is free of any major task events that might influence the time-scale calculations.

Reviewer 1, Response 12:

We apologize for not making this more clear. There are no trial-related events during this period. It is the inter-trial-interval and is devoid of any task-related information. For clarity, we have added the following text to the corresponding section of the Results:

The neural activity used to estimate the intrinsic timescales came from the last two seconds of the intertrial interval; that is, before the onset of the first offer on the next trial (similar to Murray et al., 2014). This period is absent of any cues or information about either the previous or pending trial.

Reviewer 1, Comment 13:

Lines 566-568, describing the decoding analysis, are a little confusing. For example does “1st offer and second” mean “whether the 1st or 2nd offer was chosen”? A little clarity here would be helpful.

Reviewer 1, Response 13:

We have amended the segment of the methods section as follows:

Then, we separated the data set for each neuron by the given variable label (choice of 1st or 2nd offer; choice of left or right offer; offer 1 position left or right; offer 2 position left or right; offer 1 value higher or lower than the mean value; and offer 2 value higher or lower than the mean value).

Reviewer 1, Comment 14:

What procedures were used for spike sorting?

Reviewer 1, Response 14:

For clarity, we added the following statement to the methods section:

All cells were hand-sorted using Plexon OLS.

Reviewer 1, Comment 15:

In Figure 3B-E, it's unclear why so much data is shown after ~2s; I would suggest omitting these data, so that the more informative section of the graph (~0-2.5s) is relatively larger and easier to parse.

Reviewer 1, Response 15:

Figure 3B-E have been zoomed in to scale the x-axis from -1s to 2.5s as shown here. (Note these now appear in Figure 2).

Reviewer 1, Comment 16:

In Figure 4A, I would recommend adding shading or dotted lines to indicate the major trial epochs, as in Figure 3B-E.

Reviewer 1, Response 16:

Figure 4A has been modified to reflect these recommendations. See below. (Note that this is now Figure 3.)

Reviewer 1, Comment 17:

In Figure 5, both the X and Y axes could be 'zoomed in' to give a more informative view of the data.

Reviewer 1, Response 17:

Revised as requested.

Reviewer 1, Comment 18:

In Figure 7, the alphabetical labels (A-H) could be shifted to the right, to better align with the center of each bar cluster.

Reviewer 1, Response 18:

Revised as requested

Reviewer #2 (Remarks to the Author):

Reviewer 2, Comment 1:

The term 'hierarchy', frequently used throughout the manuscript, implies a sequence, an ordered set of operations. Given that response latencies are identical in the investigated areas and that the authors do not present functional connectivity analyses I think there is little evidence to support this rather strong term.

I would suggest to use 'gradient' instead (as already done in a few instances), which is more neutral.

Reviewer 2, Response 1:

We thank the reviewer for this insightful comment. Indeed, reviewer 1 makes the same point. We now use the term gradient throughout. We have also added the following new paragraph to the Discussion that explains our terminology explicitly:

We propose that there are essentially three primary organizations the system could take: (1) modular, (2) hierarchical, and (3) a continuum/gradient. In a modular organization, each structure would be responsible for performing some unique and distinct set of computations. The differences between structures would be qualitative. That is, the type of information encoded would differ, but there would not necessarily be differences in the amount of information encoded or the speed at which it is encoded. In a hierarchical organization, information encoding would differ both quantitatively and qualitative between structures. By contrast, in a gradient organization information encoded by distinct structure would differ primarily only in the amount of encoded information of a given type. There is a general view in neuroeconomics that early reward representations encoded offers and value and later ones encode choice (Cai and Padoa-Schioppa, 2014). While there is a large body of empirical work supporting that view, our results are in keeping with several studies, included some of our own, that argue against it. In particular, in previous studies we have demonstrated robust spatial selectivity in putative early reward regions, an indication of an action plan involving the spatial orientation of the intended choice. In keeping with these previous findings, our results support an alternative view. Rather than a goods-to-action transformation, information is the same earlier in the functional gradient as it is later, which is more in-line with a cognitive map that tracks choices and potential outcomes.

Reviewer 2, Comment 2:

Brain area numbering should be included in Fig. 2B; consider showing this figure earlier since it is necessary for the non-experts to understand the neuroanatomical backdrop of the study

Reviewer 2, Response 2:

We appreciate this suggestion for improving the readability of this manuscript. We have consolidated Figures 1 and 2 into a single figure

Reviewer 2, Comment 3:

Figure 3B-E should be complemented by a population response (including a mentioning of the total number of neurons per panel)

Reviewer 2, Response 3:

We now indicate the number of cells recorded from each structure. We also include a population response, plotted as the time-resolved average firing rate across all neurons. This has been added to Figure 2 (panels F-I).

Reviewer 2, Comment 4:

The present analysis of differing selectivity to various economic variables (Fig. 4) is a little contorted and could be improved.

Reviewer 2, Response 4:

It is not clear to us what the reviewer is asking for here, although we would be happy to add any complementary analyses. Briefly, these analyses are ones that we developed in 2014 and have been using consistently since then (including in Strait et al. 2015, and in Azab and Hayden, 2017, 2018, and 2020). Including them allows for direct comparison with the past work. Moreover, we believe that these specific analyses provide a direct measure of the variables we are most interested in, as explained in detail in those papers. In any case, we are happy to improve it given specific recommendations; as it is, we think these analyses are perfectly fit to our questions.

Reviewer 2, Comment 5:

I would suggest to replace the 'proportion of neurons with selective firing rates' with the measure of explained variance. The EV can be plotted in a time-resolved manner for all variables of interest and for all brain regions. I think this would allow the reader to more quickly pick up the role of the analysed brain regions in coding particular elements of value-based reasoning.

Reviewer 2, Response 5:

We now include the suggested analyses.

Specifically, one approach to computing explained variance is to compute the Coefficient of Determination (R^2) from a regression analysis (Renaud and Victoria-Feser, 2010), which essentially equates to the squared correlation coefficient. For our modulation rate analysis, we had already computed the correlation between average firing rates and offer value. For each neuron, we computed the squared value of each correlation coefficient. We then averaged this variance coefficient across all neurons. As a percentage score, this variance coefficient thus indicates the percent of variance in average firing rate explained by offer value. We have plotted these data in both a summary and time resolved manner (new Figure 3).

Reviewer 2, Comment 6:

The layout and graphical design of the panels with time-resolved activity in Fig. 3 and Fig. 4 should be the same (i.e. choice of dashed or solid lines, marked task epochs etc)

Reviewer 2, Response 6:

These figures have now been modified to meet the recommendations of both reviewers. These modified figures have been included in this document above (see **Reviewer 2, Response 3** and **Reviewer 2, Response 5**).

Reviewer 2, Comment 7:

I. 304: should this read eight variables instead of seven?

Reviewer 2, Response 7:

This analysis has now been modified in response to comments from reviewer #1. There are 10 variables being decoded, 2 of which are decoded from firing rates in two different epochs. Thus, there are 12 independent decoding analyses. The text has been modified for clarity:

We hypothesized that the accuracy with which expected value and choice can be decoded from firing rate patterns should increase along the observed anatomical gradient. To test this hypothesis, we trained (and cross-validated) a linear classifier to decode ten binary labels (specifically: high/low expected values for *offer 1* and for *offer 2*, the *difference* between expected values of the two offers, *offer position*, *choice*, and *chosen offer position*, *chosen offer value*, *unchosen offer value*, *choice on the previous trial*, and *reward on the previous trial*) from firing rates. We investigated decoding accuracy for two of these variables (*expected value of offer 1* and *chosen offer position*) in two different epochs, thereby yielding a total of 12 independent decoding analyses (see **Methods**).

REVIEWER COMMENTS

Reviewer #1 (Remarks to the Author):

In this revised manuscript by Maisson and colleagues (from the Zimmerman lab), the authors examine the function of four cortical regions in and near the medial wall of the prefrontal cortex in monkeys performing two structurally similar decision tasks. The high level claim, updated from the previous version, is that the four regions exhibit a functional gradient, with an overall increase in decodability of task variables in dorsal as compared to ventral regions.

I am sincerely grateful to the authors for their thoughtful consideration of my prior comments, and especially for the additional analyses that they performed. While many of my concerns were satisfied, there are a few instances where the revisions fall short. In addition, the new data and figures have introduced additional concerns, both at a technical level and at a conceptual level, which are detailed below.

On the whole, I think that this paper is a worthy addition to the scientific record, and argues for a conceptual framework that should be taken seriously - specifically the idea of a non-modular organization of value- and decision-related computations within the primate frontal lobe. That said, the remaining issues significantly dampen my enthusiasm for the manuscript. In my opinion, this study, as it is presented here, represents an incremental advance, rather than a breakthrough.

1) I am convinced that on the whole there is a functional gradient with respect to the decodability of value-and choice related variables among the four regions tested. The basis for this conclusion is clear from Figure 6M. While this is potentially an important conceptual advance, the down side of such a broad generalization is that it obscures the more nuanced functional features that may exist within this overarching functional gradient. A close examination of Figure 6 illustrates what I mean: It is striking that for panels A and B (ev1-epoch1 and ev2-epoch2) vmPFC and sgACC both have very low decoding, whereas the other two regions have higher decoding. Yes, in both panels the overall gradient is present; but in addition to the gradient there appears to be a clear functional divide between the first two regions and the second two. In contrast, panels E and H (choice and previous choice during choice epoch) show a different pattern: vmPFC shows the lowest decoding by far, and the other three regions are very similar to one another. Finally, in panels J-K (all variables related to spatial direction), dACC has higher encoding accuracy than for any other variable in any other region, well above 75% for all three panels.

What this suggests to me is that within the broader gradient there is significant functional modularity. The discussion section acknowledges this, but only in reference to prior work. In my opinion, the fact that these data are being used for the exclusive purpose of illustrating one particular functional gradient is somewhat of a missed opportunity. It seems that the data could have been used to both identify this global gradient, as well as to quantify various degrees of specialization that exist among these regions. It may be the case that the gradient the authors have identified is *the* dominant feature that defines the organization of these regions, and that the examples I've cited above are only spurious observations. Or it may be that these regions are best understood as having some degree of specialization within a larger gradient. In either case, as it is now this study only tells one part of the story.

2) While I'm grateful to the authors for performing additional analyses on the response latency, this part of the paper continues to be unsatisfying. This may be due to a miscommunication on my part in my original review, and if so I apologize. What I was hoping to see was the latency to the *first change in firing rate* following the offer onsets. This is much different than the new latency analysis detailed in lines 157-163.

3) I also appreciate the inclusion of population-averaged data in Figure 2F-I. These graphs raise additional issues related to the latency analysis. It is clear from the y-axes in Figure 2F-I that the average firing rates show virtually no differentiation between the two conditions (whether offer value 1 was greater than 2, or 2 was greater than 1). This suggests that these averages largely

reflect the activity of neurons that have little or no task modulation within this epoch, and therefore that the latency measurements largely reflect the firing of non-modulated cells. (This is consistent with Figure 3, which shows that only a small fraction of cells is modulated by task variables.) This suggests the possibility that no difference in latencies was found because most of the cells in this analysis are simply un-responsive to the task events. Until this question can be resolved, I don't think the latency results shown in Figure 2A can be considered reliable.

4) I'd like to offer a suggestion on how to resolve the two points above. First, there are two benchmarks that are potentially of interest with respect to latency: changes in firing rate, and the emergence of encoding of task-relevant variables. These two often occur at the same time, but they need not. In other words, one can imagine a neuron that changes its firing rate in response to stimulus onset, but does not encode any information about that stimulus (e.g. value). So, one way forward here would be to analyze the latency in terms of both of these benchmarks. For change in firing rate, all of the task conditions could be collapsed, so that what is being measured is the global change in firing rate regardless of values, choices, etc. Here, I think it would be wise to separately identify the neurons with firing rate increases and firing rate decreases; and to attempt to identify the latency to the first change in firing rate as well as the latency to peak change in firing rate. Neurons that seem to have no change in firing rate would be excluded. Likewise, for the encoding of task variables, I think it would be informative to identify the first significant bin (or string of significant bins), as well as the bin at which the effect size reaches its peak. Neurons with no significant encoding would be excluded from this analysis. Together, this would produce a more comprehensive picture of when neurons in these areas begin to be sensitive to task events, and when they begin to encode important task information. Moreover, excluding non-responsive and non-coding cells would avoid the potential for artifactual negative findings.

5) My thanks to the authors for including in their reply additional data regarding firing rates, fano factor, and intrinsic dimensionality. Why not include this in the manuscript? I think that a more comprehensive reporting of these basic properties would add significantly to the contribution this paper makes.

6) That said, I am deeply confused about the fano factors reported in the author's reply. My understanding is that the fano factor is defined as the ratio of the firing rate variance to the firing rate means. Because variances and mean firing rates are always positive, I don't understand how *negative* fano factors could be reported in the reply. (The reply cites Chang et al. 2012, which I'm assuming is Chang, Armstrong, and Moore, J Neurosci 32:2204; this paper only reports positive fano factors, e.g. their Figure 3B.) If there is some other common definition of fano factor that can take negative values, then I apologize. In any case, some clarification here is warranted.

7) Minor comment: In Figure 3, is the smoothing different for panels A-D as compared to F-I? My instinct is that these should reflect the same level of smoothing, so I'm curious why it appears that they do not.

Reviewer #2 (Remarks to the Author):

I would like to commend the authors for their extensive and thorough revisions. They have addressed all of my main concerns. Before the manuscript is published, I suggest to add a measure of variance to the new graphs with the population analyses in Figs. 2 & 3 (F-I), e.g. SEM, and to normalize unit firing rates in Fig. 2 before averaging, e.g. by z-scoring.

Reviewer #1 (Remarks to the Author):

I am sincerely grateful to the authors for their thoughtful consideration of my prior comments, and especially for the additional analyses that they performed. While many of my concerns were satisfied, there are a few instances where the revisions fall short. In addition, the new data and figures have introduced additional concerns, both at a technical level and at a conceptual level, which are detailed below. On the whole, I think that this paper is a worthy addition to the scientific record, and argues for a conceptual framework that should be taken seriously - specifically the idea of a non-modular organization of value- and decision-related computations within the primate frontal lobe. That said, the remaining issues significantly dampen my enthusiasm for the manuscript.

Reviewer 1, Comment 1:

I am convinced that on the whole there is a functional gradient with respect to the decode-ability of value and choice related variables among the four regions tested. The basis for this conclusion is clear from Figure 6M. While this is potentially an important conceptual advance, the down side of such a broad generalization is that it obscures the more nuanced functional features that may exist within this overarching functional gradient. A close examination of Figure 6 illustrates what I mean: It is striking that for panels A and B (ev1-epoch1 and ev2-epoch2) vmPFC and sgACC both have very low decoding, whereas the other two regions have higher decoding. Yes, in both panels the overall gradient is present; but in addition to the gradient there appears to be a clear functional divide between the first two regions and the second two.

We respectfully disagree that the decoding in A and B is “very low”. That’s a judgment call - interpreting that specific number as “very low” requires an assumption about what reasonable decoding would be and that, in turn, would need to incorporate information about cell selection criteria.

More generally, it is not appropriate to draw conclusions about the relative lengths of the bars as the reviewer does, because doing so makes an unwarranted and likely problematic assumption about the *linearity* of the decodability dimension plotted on the y-axis. That is, we cannot adjudge that the difference in decodability between 60% to 65% is comparable to the difference in decodability between 65% and 70%. Formally speaking, all we can conclude from the data is that decoding is significantly above chance in all areas, and that this effect increases with region along our proposed hierarchy.

To use the reviewer’s first example, to conclude that vmPFC and sgACC are part of a module based on a failure to find a difference between their firing while arguing that there is a modular difference between this pair and the other two regions, is precisely the statistical mistake warned against in this paper:

Nieuwenhuis, S., Forstmann, B. U., & Wagenmakers, E. J. (2011). Erroneous analyses of interactions in neuroscience: a problem of significance. *Nature neuroscience*, 14(9), 1105.

In any case, it is always tempting to eyeball the data and try to tell a story based on the patterns, but doing so risks apophenia. That’s why we favor the most

conservative hypothesis testing statistical approach, which is to limit ourselves to our specific a priori hypotheses.

In contrast, panels E and H (choice and previous choice during choice epoch) show a different pattern: vmPFC shows the lowest decoding by far, and the other three regions are very similar to one another. Finally, in panels J-K (all variables related to spatial direction), dACC has higher encoding accuracy than for any other variable in any other region, well above 75% for all three panels. What this suggests to me is that within the broader gradient there is significant functional modularity.

More broadly, we think the reviewer is using the term modularity in a different sense than us. We are using it in the sense that a neuroimager or someone using lesion studies would - that each area has a specific unique intrinsic function. That definition would predict that for the functions we identify - corresponding to the panels on figure 6 - one region would have much greater decoding than the others and that would not be the hierarchically last region on each graph.

One aspect of the reviewer's proposal seems to be that there may be a step-like organization where there is a qualitative leap between areas in a series, even though the function is found in increasing amounts along each area. Our data are consistent with this possibility, although we have no way to exclude the possibility that there are other intervening regions that would make the steps smoother. However, we would definitely not use the term modular for this organization; we would use the term gradient. We have added the following text to the Discussion highlighting this fact.

One possibility is that there is a modular organization of a somewhat different form than we have proposed. Specifically, it is possible that each region in the series represents a large qualitative step from the one before. Unfortunately, our data are not sufficient to differentiate this organization from smoother ones. Either way, our data are consistent with a series in which each area carries information in a progressively more legible format.

The discussion section acknowledges this, but only in reference to prior work. In my opinion, the fact that these data are being used for the exclusive purpose of illustrating one particular functional gradient is somewhat of a missed opportunity. It seems that the data could have been used to both identify this global gradient, as well as to quantify various degrees of specialization that exist among these regions.

This is a point where we simply disagree with the reviewer. There is an increase in these regions by all measures. Consequently, the only conclusion that can be made about degrees of specialization is that the hierarchically last region (the dACC) is the most specialized for all functions (or at least, has the most decodable representation). That would seem to be evidence against any reasonable definition of modular architecture. We hope that our additional text on modularity clarifies our position.

It may be the case that the gradient the authors have identified is *the* dominant feature that defines the organization of these regions, and that the examples I've cited above are only spurious observations. Or it may be that these regions are best understood as having some

degree of specialization within a larger gradient. In either case, as it is now this study only tells one part of the story.

Reviewer1, Comment 2:

While I'm grateful to the authors for performing additional analyses on the response latency, this part of the paper continues to be unsatisfying. This may be due to a miscommunication on my part in my original review, and if so I apologize. What I was hoping to see was the latency to the *first change in firing rate* following the offer onsets. This is much different than the new latency analysis detailed in lines 157-163.

Reviewer 1, Response 2:

We thank Reviewer 1 for providing this additional clarification, and we apologize sincerely for misunderstanding the requested analysis. We have added an additional latency analysis, characterizing the average time elapsed from the offer onset to the the first significant change in firing rate. To do this, we calculated the correlations between mean firing rates and the expected value of offer 1 in epoch 1, and of offer 2 in epoch 2. We determined the time at which this correlation became statistically significant, and computed the time in ms from the onset of the offer and this significance threshold. As recommended by the reviewer, we eliminated neurons for which not significant correlation between firing rate and expected value was evident. We then averaged latency across all neurons and used a 4 (area) x 2 (offer 1 or 2) ANOVA to test for differences. Neither the main effect of area ($F = 1.14$, $p = 0.457$) nor the offer number ($F = 0.041$, $p = 0.853$) was statistically significant. We repeated this analysis without removing any neurons. The results remained unchanged. In conjunction with the previous revision, in which we calculated the latency between the significance threshold and peak firing, we believe this constitutes the second of the reviewers 2 recommendations in Comment 4. We have added the following text to the results section of the manuscript:

Using this definition of a significance threshold, we also computed the time elapsed from the onset of the offer to the first time point at which firing rates were significantly correlated with expected value. We averaged latency across neurons and used a 4 (area) x 2 (offer 1 or 2) ANOVA to test for differences. Neither the main effect of area ($F = 1.14$, $p = 0.457$) nor the offer number ($F = 0.041$, $p = 0.853$) was statistically significant. We repeated this analysis without removing any neurons. The results remained unchanged.

To follow Reviewer 1's second recommendation, we calculated latency as the time elapsed from the onset of an offer (in each epoch) to both the peak change in firing rate and to the first change in firing rate. To do this, we calculated the mean firing rate for each neuron across trials. We then calculated both the mean and standard deviation in firing rate across each epoch. Next, for each time bin, we calculate the absolute value of the difference between firing rate and mean firing rate. We then identified the time bin of the peak change in firing rate relative to the mean firing rate for each neuron. For each epoch, we calculated the time elapsed from the onset of the offer to the peak change in firing rate and averaged these latencies across neurons. We performed a 4 (area) x 2 (offer 1 or 2) ANOVA to look for differences in time elapsed to peak change in firing rate. For both the main

effect of area ($F = 0.44$, $p = 0.742$) and the main effect of offer ($F = 3.84$, $p = 0.145$) we found no significant differences.

We then defined the time of the first significant change in firing rate as the first time at which firing rates were either greater than 2 standard deviations above or below (to account for neurons with firing rates that decrease) the mean. For each neuron, we then calculated the time elapsed between the onset of the offer and this first significant change in firing rate. For each epoch, we averaged latencies across neurons and performed a 4 (area) x 2 (offer 1 or 2) ANOVA to look for differences in time elapsed to the first significant change in firing rate. For both the main effect of area ($F = 0.76$, $p = 0.448$) and the main effect of offer ($F = 0.17$, $p = 0.912$) we found no significant differences. We have added the following text to the results section of the manuscript:

Finally, we also calculated latency as the time elapsed from the onset of an offer (in each epoch) to both the peak change in firing rate and to the first significant change in firing rate, irrespective of variable encoding. To do this, we calculated the mean firing rate for each neuron across trials. We then calculated both the mean and standard deviation in firing rate across each epoch. Next, for each time bin, we calculate the absolute value of the difference between firing rate and mean firing rate. We then identified the time bin of the peak change in firing rate relative to the mean firing rate for each neuron. For each epoch, we calculated the time elapsed from the onset of the offer to the peak change in firing rate and averaged these latencies across neurons. We performed a 4 (area) x 2 (offer 1 or 2) ANOVA to look for differences in time elapsed to peak change in firing rate. For both the main effect of area ($F = 0.44$, $p = 0.742$) and the main effect of offer ($F = 3.84$, $p = 0.145$) we found no significant differences. We then defined the time of the first significant change in firing rate as the first time at which firing rates were either greater than 2 standard deviations above or below (to account for neurons with firing rates that decrease) the mean. For each neuron, we then calculated the time elapsed between the onset of the offer and this first significant change in firing rate. For each epoch, we averaged latencies across neurons and performed a 4 (area) x 2 (offer 1 or 2) ANOVA to look for differences in time elapsed to the first significant change in firing rate. For both the main effect of area ($F = 0.76$, $p = 0.448$) and the main effect of offer ($F = 0.17$, $p = 0.912$) we found no significant differences.

Reviewer 1, Comment 3:

I also appreciate the inclusion of population-averaged data in Figure 2F-I. These graphs raise additional issues related to the latency analysis. It is clear from the y-axes in Figure 2F-I that the average firing rates show virtually no differentiation between the two conditions (whether offer value 1 was greater than 2, or 2 was greater than 1). This suggests that these averages largely reflect the activity of neurons that have little or no task modulation within this epoch, and therefore that the latency measurements largely reflect the firing of non-modulated cells. (This is consistent with Figure 3, which shows that only a small fraction of cells is modulated by task variables.) This suggests the possibility that no difference in latencies was found because most of the cells in this analysis are simply unresponsive to the task events. Until this question can be resolved, I don't think the latency results shown in Figure 2A can be considered reliable.

Reviewer 1, Response 3:

This comment raises a valid point and it is for this reason that we restricted the latency analysis (above) to significantly modulated neurons.

Reviewer 1, Comment 4 (Recommendations for Comment 2 and 3):

I'd like to offer a suggestion on how to resolve the two points above. First, there are two benchmarks that are potentially of interest with respect to latency: changes in firing rate, and the emergence of encoding of task-relevant variables. These two often occur at the same time, but they need not. In other words, one can imagine a neuron that changes its firing rate in response to stimulus onset, but does not encode any information about that stimulus (e.g. value). So, one way forward here would be to analyze the latency in terms of both of these benchmarks.

- 1) For change in firing rate, all of the task conditions could be collapsed, so that what is being measured is the global change in firing rate regardless of values, choices, etc. Here, I think it would be wise to separately identify the neurons with firing rate increases and firing rate decreases; and to attempt to identify the latency to the first change in firing rate as well as the latency to peak change in firing rate. Neurons that seem to have no change in firing rate would be excluded.
- 2) Likewise, for the encoding of task variables, I think it would be informative to identify the first significant bin (or string of significant bins), as well as the bin at which the effect size reaches its peak. Neurons with no significant encoding would be excluded from this analysis. Together, this would produce a more comprehensive picture of when neurons in these areas begin to be sensitive to task events, and when they begin to encode important task information. Moreover, excluding non-responsive and non-coding cells would avoid the potential for artifactual negative findings.

Reviewer 1, Response 4:

We thank the reviewer for the comments provided and the time spent on significantly improving the manuscript as well as our analysis. Revisions are included in the responses above. In short, we find that, no matter what analysis we do, we still do not see any latency effects in the data. This is, of course, a null finding, and so we cannot draw strong conclusions from it, as we indicate in our manuscript.

Reviewer 1, Comment 5:

My thanks to the authors for including in their reply additional data regarding firing rates, fano factor, and intrinsic dimensionality. Why not include this in the manuscript? I think that a more comprehensive reporting of these basic properties would add significantly to the contribution this paper makes.

Reviewer 1, Response 5:

We agree with this assessment and have added the following paragraphs to the results section.

Baseline firing rates do not differ between structures

First, we asked whether there were any differences in the intrinsic firing properties of neurons between the target structures. To measure baseline firing rates in each structure, we computed the average firing rate for each neuron during the 1-second pre-trial period, from -2 s to 0 s, where time 0 was set to the onset of offer 1. (Note that this is the same time window on which we performed our analysis of intrinsic timescales). The baseline firing rate in vmPFC was 3.11 spikes/second. The baseline firing rate in sgACC was 3.69 spikes/second. The baseline firing rate in pgACC was 0.75 spikes/second. Finally, the baseline firing rate in dACC was 5.06 spikes/second. A

Spearman correlation of these baseline firing rates with the proposed order was not significant ($\rho = 0.4$, $p = 0.75$, Spearman correlation). As the results are non-significant, we cannot draw any additional conclusions from this analysis.

Next, we wanted to determine if variability in the firing rates differed between structures. To measure firing rate variability, we computed the average Fano Factor (FF) across neurons in each region, replicating the procedure outlined in Chang et al. (2012). For each time bin (20 ms) in each epoch (500 ms), we computed the average firing rate (FR) from across trials. We then calculated the variance and mean for firing rates across all time bins, within the given epoch. For each cell, the FF was calculated as the ratio of firing rate variance to average firing rate across the epoch. We then calculated the average FF across cells. As a result, in epoch 1: vmPFC FF = 0.0024, sgACC FF = 0.1411, pgACC FF = 0.0025, dACC FF = 0.1756; in epoch 2: vmPFC FF = 0.0025, sgACC FF = 0.1422, pgACC FF = 0.0025, dACC FF = 0.1180. We computed a Spearman correlation between the FF and the proposed gradient order, for each epoch. In epoch 1, the correlation was not significant ($\rho = 0.8$, $p = 0.33$). In epoch 2, the correlation was again not significant ($\rho = 0.4$, $p = 0.75$). As the results are non-significant, we cannot draw any additional conclusions from this analysis.

Lastly, we reasoned that qualitative encoding differences between structures would be associated with differences in the degree of dimensionality in neural activity between structures. To measure intrinsic dimensionality, we performed a principal component analysis on the normalized firing rates across neurons in each of the offer epochs, independently for each structure. For each neuron, we isolated the firing rates from both offer epochs. We calculated the mean firing rate across trials for each 20 ms time bin. Using these average responses, we composed a matrix of time X neurons. We performed an eigenvalue decomposition and computed the explained variance due to each of the first three principal components (PCs). We then calculated the degree of change, by calculating the slope, in explained variance from the first PC to the second PC, and again from the second PC to the third. We performed this same analysis for each offer epoch independently. Finally, we asked if the change in explained variance between the first three PCs was significantly correlated with the proposed order. We found that in epoch 1, the change in explained variance between PC1 and PC2 did not significantly change with gradient order ($\rho = -1$, $p = 0.083$, Spearman correlation). Similarly, the change in explained variance between PC2 and PC3, in epoch 1, did not significantly change with gradient order ($\rho = -1$, $p = 0.083$, Spearman correlation). We found that in epoch 2, the change in explained variance between PC1 and PC2 did not significantly change with gradient order ($\rho = -1$, $p = 0.083$, Spearman correlation). The change in explained variance between PC2 and PC3, in epoch 2, did not significantly decrease with gradient order ($\rho = -0.4$, $p = 0.75$, Spearman correlation). As the results are not significant, in any of these three analyses of intrinsic properties, we were unable to identify any qualitative differences in signatures of neural activity between the target structures.

Reviewer 1, Comment 6:

That said, I am deeply confused about the fano factors reported in the author's reply. My understanding is that the fano factor is defined as the ratio of the firing rate variance to the firing rate means. Because variances and mean firing rates are always positive, I don't understand how *negative* fano factors could be reported in the reply. (The reply cites Chang et al. 2012, which I'm assuming is Chang, Armstrong, and Moore, J Neurosci 32:2204; this paper only reports positive fano factors, e.g. their Figure 3B.) If there is some other common definition of

fano factor that can take negative values, then I apologize. In any case, some clarification here is warranted.

Reviewer 1, Response 6:

We thank the reviewer for catching this error. We carelessly performed the Fano Factor analysis on our z-scored firing rates, rather than on our raw firing rates. We have made this correction to the manuscript and apologize. In short, the results don't change when we do the analysis the correct way.

To measure firing rate variability, we computed the average Fano Factor (FF) across neurons in each region, replicating the procedure outlined in Chang et al. (2012). For each time bin (20 ms) in each epoch (500 ms), we computed the average firing rate (FR) from across trials. We then calculated the variance and mean for firing rates across all time bins, within the given epoch. For each cell, the FF was calculated as the ratio of firing rate variance to average firing rate across the epoch. We then calculated the average FF across cells. As a result, in epoch 1: vmPFC FF = 0.0024, sgACC FF = 0.1411, pgACC FF = 0.0025, dACC FF = 0.1756; in epoch 2: vmPFC FF = 0.0025, sgACC FF = 0.1422, pgACC FF = 0.0025, dACC FF = 0.1180. We computed a Spearman correlation between the FF and the proposed gradient order, for each epoch. In epoch 1, the correlation was not significant ($\rho = 0.8$, $p = 0.33$). In epoch 2, the correlation was again not significant ($\rho = 0.4$, $p = 0.75$). As the results are non-significant, we cannot draw any additional conclusions from this analysis.

Reviewer 1, Comment 7:

Minor comment: In Figure 3, is the smoothing different for panels A-D as compared to F-I? My instinct is that these should reflect the same level of smoothing, so I'm curious why it appears that they do not.

Reviewer 1, Response 7:

We thank the reviewer for pointing out this discrepancy in our plotting. We did erroneously apply a smoothing filter to the explained variance traces, which was not applied to the modulation rate traces. This has been corrected in the corresponding Figure, shown below.

Reviewer #2 (Remarks to the Author):

Reviewer 2, Comment 1:

Before the manuscript is published, I suggest to add a measure of variance to the new graphs with the population analyses in Figs. 2 & 3 (F-I), e.g. SEM, and to normalize unit firing rates in Fig. 2 before averaging, e.g. by z-scoring.

Reviewer 2, Response 1:

We thank the reviewer for this additional feedback. We have modified both Figs. 2 & 3 according to these recommendations. We appreciate the opportunity to raise the quality of the manuscript.

Figure 2

Figure 3

REVIEWER COMMENTS

Reviewer #1 (Remarks to the Author):

My thanks to the authors for responding to my previous comments. My remarks on this revision will be brief.

With respect to gradients vs. modules, and the interpretation of the results: I absolutely agree that eye-balling the bar graphs is not the right way to assess the data, and that some quantitative, hypothesis-based approach is the way to go. The point of my earlier comment was to urge the authors to quantitatively test hypotheses other than the most basic one of a single gradient. This might include the possibility of step-like transitions for some of the variables, for example. I think it would give a more complete explanation of the pattern of decoding across these four regions.

Fano factor: My thanks again for re-doing this analysis. I appreciate that the authors are willing to entertain this request. At this stage, it's a very minor point relative to the rest of the paper, but I'm afraid that the results still seem to be somewhat unusual, and need to be reexamined: For cortical neurons, the fano factor is usually around 1, or slightly higher. See for example Chang, Armstrong, and Moore (2012) and also Figure 3 of Churchland et al. (Nat. Neuro 13:369, 2010). The expectation of $FF \sim 1$ is supported by the notion that spikes are generated by a Poisson-like process, such that the number of spikes observed from a single neuron over a given time span is approximately Poisson-distributed, i.e. with the variance equal to the mean. In empirical measurements of single neuron activity, the spike counts are usually over-dispersed, with variances slightly higher than the mean, producing $FF > 1$. In contrast, in the updated results none of the fano factors are even near 1, and some are as low as 0.0024. To achieve a fano-factor this low, a neuron would have to have a constant, pacemaker-like train of spikes (i.e. extremely low variance in spike counts), very unlike what actual cortical neurons do. In any case, this result is very unusual, and suggests the possibility of another oversight during the analysis. I would urge the authors to get to the bottom of this strange result, and to double-check any other analyses that could potentially be affected.

Small comment: The four colors used to indicate the different brain areas are all nearly isoluminant and equally saturated, and are therefore unlikely to be friendly to colorblind readers. Please consider updating the colors so that they can be distinguished by all readers.

Reviewer 1, Comment 1:

With respect to gradients vs. modules, and the interpretation of the results: I absolutely agree that eye-balling the bar graphs is not the right way to assess the data, and that some quantitative, hypothesis-based approach is the way to go. The point of my earlier comment was to urge the authors to quantitatively test hypotheses other than the most basic one of a single gradient. This might include the possibility of step-like transitions for some of the variables, for example. I think it would give a more complete explanation of the pattern of decoding across these four regions.

Reviewer 1, Response 1:

We appreciate the reviewer's patience in re-evaluating our manuscript. However, we remain unable to address this concern fully. To briefly explain our rationale:

1. Distinguishing stepping from ramping is really hard

The reviewer specifically mentions stepping and ramping as possibilities. Distinguishing these possibilities is extremely difficult. Recall that there is a well-known debate in the LIP literature that relates to the much more tractable problem of ramping vs stepping in the dynamics of evidence accumulation in LIP neurons. We are not involved in that argument, nor have we taken sides in it, but we have had detailed discussions with key players of both sides. One important lesson that can be learned from the length and acrimony of the debate is that devising statistical tests to distinguish ramping from stepping in serial data is quite difficult unless the pattern is overwhelmingly obvious. Simple visual inspection of our data demonstrates that ours is not one of those obvious cases. (In our case, the problem is even harder than it is in LIP because the independent variable is not linear and because we have a limited x-axis, only four regions). To be more precise, we believe that we *could* devise a test that disambiguates ramping from stepping, but that would require making many assumptions, and the answer we could obtain would basically result from those assumptions, not from the data, so it would not illuminate the question the reviewer is asking.

2. Because we only recorded four areas, our data cannot, even in principle, answer this question.

The reviewer agrees with our central claim of a monotonic increase in decodability across areas. The reviewer is then asking a secondary question, namely, the nature of that monotonic relationship. We agree that this is a valid question, however, we do not have the right data to answer it. The reason is that to answer this question, we would need to sample evenly across the extent of the medial PFC, and across all the major layers of cortex (since each layer likely engages in somewhat distinct computations). We did not do that - instead, we sampled from four specific regions out of more than a dozen, and combined data across layers. To sample evenly would take, we estimate, at least 10x as much data as we collected. (And note that, with four areas, ours is already a relatively large dataset). The reason this is important is that, in the absence of a full connectivity profile of the medial PFC (and no such map exists), we cannot know whether apparent steps would actually be smooth slopes if we had the intermediate areas. But we currently do not know if there are intermediate areas, or how many they are. Thus, even if we devised a statistical procedure to disambiguate steps

from slopes, we would not be able to draw conclusions from the results of that procedure. Any such conclusions would be artifacts of the brain regions we happened to sample.

3. The reviewer's comment is not relevant to our hypotheses

The main claim of the paper is that areas have a gradient organization, not a modular structure. (Our results also argue for a specific serial organization, although the reviewer accepts our proposed series, so that part is not germane to this review concern). This gradient organization could come in either of two flavors, continuum or step-like. Which of these describes the organization of the brain is an interesting question, but it is fundamentally a smaller one - that is, it does not tie into major debates about cortical organization, nor does it relate to macro-level questions. It is also not the question we set out to study, nor one our data can be used to answer (see point 2 above). A modular organization would require a degree of specialization, which would show up in the form of each variable being associated most strongly with one of the four areas. We plainly do not see that, so we can reject the modular hypothesis.

In summary, we feel that the reviewer's question is both interesting and valid. However, it is not germane to our scientific interests, would require an order of magnitude more data to answer, and would necessarily rely on some very tricky statistical techniques, which may not exist. At the same time, the reviewer's question indicates that they accept our major claims - a specific gradient - meaning that the paper does achieve its goals.

Because the reviewer's thoughts may resemble those of future readers, we summarize these ideas in a new paragraph we have added to the Discussion, as follows:

Our results demonstrate a clear monotonic organization of decodability across areas. One important question that our results do not address is the specific form of this monotonic organization. For example, it is possible that the processing occurs in a smooth and gradual form; another possibility is that it occurs in a series of steps; indeed, our data cannot reject the hypothesis that it is steplike for some variables and smooth for others. Successful adjudication between these two hypotheses would require two things, (1) much denser sampling of the medial prefrontal cortex, including medial prefrontal areas that we did not sample at all and sampling across all layers of cortex, and (2) the development of novel statistical techniques that can unambiguously dissociated ramping from stepping. We are optimistic that new recording technologies will enable (1) and that the existence of data from (1) will motivate that development of (2). In any case, the specific structure of the organization of change across areas remains an open question.

Reviewer 1, Comment 2:

Fano factor: My thanks again for re-doing this analysis. I appreciate that the authors are willing to entertain this request. At this stage, it's a very minor point relative to the rest of the paper, but I'm afraid that the results still seem to be somewhat unusual, and need to be reexamined: For cortical neurons, the fano factor is usually around 1, or slightly higher. See for example Chang, Armstrong, and Moore (2012) and also Figure 3 of Churchland et al. (Nat. Neuro 13:369, 2010).

The expectation of $FF \sim 1$ is supported by the notion that spikes are generated by a Poisson-like process, such that the number of spikes observed from a single neuron over a given time span is approximately Poisson-distributed, i.e. with the variance equal to the mean. In empirical measurements of single neuron activity, the spike counts are usually over-dispersed, with variances slightly higher than the mean, producing $FF > 1$. In contrast, in the updated results none of the fano factors are even near 1, and some are as low as 0.0024. To achieve a fano-factor this low, a neuron would have to have a constant, pacemaker-like train of spikes (i.e. extremely low variance in spike counts), very unlike what actual cortical neurons do. In any case, this result is very unusual, and suggests the possibility of another oversight during the analysis. I would urge the authors to get to the bottom of this strange result, and to double-check any other analyses that could potentially be affected.

Reviewer 1, Response 2:

We thank the reviewer for their commitment to ensuring the quality of these statistical results. There are many approaches to quantifying Fano Factor, which can yield slightly different results, depending on the assumptions made. In our earlier reply, we first averaged neural responses across trials before computing Fano Factor. In the re-revised text, we now first compute the Fano Factor for each trial and each neuron before averaging the results across trials and neurons. We sense this may be the more standard way (e.g. Chang et al., 2012). When we follow the procedure used in Chang et al., we find results more in line with the reviewer's expectations. We've adjusted the corresponding section of the results accordingly to include that analysis and not include the other one.

Next, we wanted to determine if variability in the firing rates differed between structures. To measure firing rate variability, we computed the average Fano Factor (FF) across neurons in each region, replicating the procedure outlined in Chang et al. (2012). In each epoch (500 ms), we computed the firing rate variance and firing rate average on each trial. We then averaged this FF across trials for each epoch and each neuron. Finally, we computed the average FF across neurons in each structure, for each epoch. As a result, in epoch 1: vmPFC FF = 1.065 (SEM: 0.0099), sgACC FF = 1.009 (SEM: 0.0062), pgACC FF = 1.001 (SEM: 0.0041), dACC FF = 1.009 (SEM: 0.0062); in epoch 2: vmPFC FF = 1.066 (SEM: 0.0098), sgACC FF = 1.008 (SEM: 0.0066), pgACC FF = 0.999 (SEM: 0.0038), dACC FF = 1.008 (SEM: 0.0066). We computed a Spearman correlation between the FF and the proposed gradient order, for each epoch. In both epoch 1 and epoch 2, the correlation was not significant ($\rho = -0.63$, $p = 0.5$). As the results are non-significant, we cannot draw any additional conclusions from this analysis.

Reviewer 1, Comment 3:

Small comment: The four colors used to indicate the different brain areas are all nearly isoluminant and equally saturated, and are therefore unlikely to be friendly to colorblind readers. Please consider updating the colors so that they can be distinguished by all readers.

Reviewer 1, Response 3:

We appreciate and share the reviewer's concern for, and commitment to, equity and accessibility. This is why in writing the paper, we deliberately used an online color blind-friendly palette generator to select these specific tones (<https://davidmathlogic.com/colorblind/>). We apologize that we neglected to mention this information in the Methods; we now include it in the revised

Methods. Thanks to the palette generator, we are confident that the difference in luminance will allow for distinguishing between structures. Having said that, if the reviewer or editor disagrees with the recommendations of that website, we are happy to use other colors.

REVIEWER COMMENTS

Reviewer #2 (Remarks to the Author):

The authors have provided an additional paragraph with further information on the nature of monotonic organization of decodability. I concur with their line of argumentation and consider this reviewer comment (#1) to have been adequately addressed.

Reviewer comment #3 (choice of colors) has been adequately addressed.

Regarding reviewer comment #2 (Fano factor), I am suspecting the authors mis-interpreted Chang et al. (2012): the FF is typically given as an across-trial measure, i.e. one determines the variance and the mean of binned spike counts across trials. Per bin and neuron, one FF is obtained. The FF is not calculated per trial and then averaged, as the authors currently do. I recommend a re-analysis of the FF that follows this convention.

Reviewer 2, Comment 1:

Regarding reviewer comment #2 (Fano factor), I am suspecting the authors mis-interpreted Chang et al. (2012): the FF is typically given as an across-trial measure, i.e. one determines the variance and the mean of binned spike counts across trials. Per bin and neuron, one FF is obtained. The FF is not calculated per trial and then averaged, as the authors currently do. I recommend a re-analysis of the FF that follows this convention.

Reviewer 1, Response 2:

We sincerely thank the reviewer for this feedback. We apologize for any inconsistencies. We have now read the paper carefully and have performed the recommended analysis. We apologize again for not having done this earlier.

Specifically, we first segmented each 500 ms epoch into 100 ms time bins. For each neuron, and in each time bin, we calculate the variance and mean across trials. We then divided the variance by the mean to compute the FF within the time bin across trials. Finally, we average across neurons to estimate the FF within the recorded population for each time bin. The following paragraph has been modified in the results section to reflect these changes:

Next, we wanted to determine if variability in the firing rates differed between structures. To measure firing rate variability, we computed the average Fano Factor (FF) across neurons in each region, replicating the procedure outlined in Chang et al. (2012)⁶⁸. We segmented each 500ms epoch into 100 ms bins. For each neuron, we calculate the variance and mean across trials within a given 100 ms bin. We then computed the Fano Factor (FF) as the variance divided by the mean and averaged this FF across neurons. Finally, in order to compare the FF across structure against the proposed hierarchy, we performed a Spearman correlation between the average FF across time bins and the expected gradient order. Across the time bins in epoch 1, the average FF in vmPFC was 1.93 ± 0.199 (SEM). The average FF in sgACC was 1.24 ± 0.023 . The average FF in pgACC was 1.23 ± 0.025 . The average FF in dACC was 1.35 ± 0.042 . (Epoch 2, vmPFC: 1.92 ± 0.171 ; sgACC: 1.24 ± 0.025 ; pgACC: 1.23 ± 0.024 ; dACC: 1.32 ± 0.042). There was no significant correlation between gradient order and FF across structure in either epoch 1 ($\rho = -0.4$, $p = 0.75$, Spearman correlation) or epoch 2 ($\rho = 0.4$, $p = 0.74$, Spearman correlation). As the results are non-significant, we cannot draw any additional conclusions from this analysis.

REVIEWERS' COMMENTS

Reviewer #2 (Remarks to the Author):

The revised calculation of the Fano factor now follows convention, i.e. it is given as an across-trial measure of variability. All comments have now been satisfactorily addressed. I would like to congratulate the authors on a very nice paper.